# Effector prediction and characterization in the oomycete pathogen *Bremia lactucae* reveal host-recognized WY domain proteins that lack the canonical RXLR motif

**Kelsey J. Wood**[1,2], **Munir Nur**[1], **Juliana Gil**[1,3], **Kyle Fletcher**[1], **Kim Lakeman**[4], **Dasan Gann**[1], **Ayumi Gothberg**[1], **Tina Khuu**[1], **Jennifer Kopetzky**[1], **Sanye Naqvi**[1], **Archana Pandya**[1], **Chi Zhang**[1], **Brigitte Maisonneuve**[5], **Mathieu Pel**[4], **Richard Michelmore**[1,6]*

1 The Genome Center, University of California, Davis, Davis, California, United States of America, 2 Integrative Genetics & Genomics Graduate Group, University of California, Davis, Davis, California, United States of America, 3 Plant Pathology Graduate Group, University of California, Davis, Davis, California, United States of America, 4 Enza Zaden, Enkhuizen, The Netherlands, 5 INRAE, UR1052, GAFL, Montfavet, France, 6 Departments of Plant Sciences, Molecular & Cellular Biology, Medical Microbiology & Immunology, University of California, Davis, Davis, California, United States of America

* rwmichelmore@ucdavis.edu

**Data Availability Statement:** The reference genome assembly of *B. lactucae* is available from NCBI, GenBank ID: GCA_004359215.1. RNAseq

## Abstract

Pathogens that infect plants and animals use a diverse arsenal of effector proteins to suppress the host immune system and promote infection. Identification of effectors in pathogen genomes is foundational to understanding mechanisms of pathogenesis, for monitoring field pathogen populations, and for breeding disease resistance. We identified candidate effectors from the lettuce downy mildew pathogen *Bremia lactucae* by searching the predicted proteome for the WY domain, a structural fold found in effectors that has been implicated in immune suppression as well as effector recognition by host resistance proteins. We predicted 55 WY domain containing proteins in the genome of *B. lactucae* and found substantial variation in both sequence and domain architecture. These candidate effectors exhibit several characteristics of pathogen effectors, including an N-terminal signal peptide, lineage specificity, and expression during infection. Unexpectedly, only a minority of *B. lactucae* WY effectors contain the canonical N-terminal RXLR motif, which is a conserved feature in the majority of cytoplasmic effectors reported in *Phytophthora* spp. Functional analysis of 21 effectors containing WY domains revealed 11 that elicited cell death on wild accessions and domesticated lettuce lines containing resistance genes, indicative of recognition of these effectors by the host immune system. Only two of the 11 recognized effectors contained the canonical RXLR motif, suggesting that there has been an evolutionary divergence in sequence motifs between genera; this has major consequences for robust effector prediction in oomycete pathogens.

reads of lettuce cv. Cobham Green infected with *B. lactucae* isolate SF5 is available at NCBI SRA BioProject PRJNA523226. Sequences of effector proteins are available in File S1. All data for interactions between effectors and individual lines of *Lactuca* spp. are available at http://bremia. ucdavis.edu/BIL/BIL_interaction.php. The sequences of ORFs from the oomycete genomes used for effector prediction, along with scripts used for their analysis, can be found on Github at https:// github.com/kelseywood/Bremia-lactucae-WY-effector-characterization.

**Funding:** The work was supported by an NSF Graduate Research Fellowship and a USDA Fellowship #2018-67011-28053 to KW and the NSF/USDA AFRI Microbial Sequencing Program award #2009-65109-05925 to RWM. The funders had no role in study design, data collection and analysis, decision to publish, or preparation of the manuscript.

**Competing interests:** The authors have declared that no competing interests exist.

## Author summary

There is a molecular battle that takes place during infection of plants and animals by pathogens. Some of the weapons in the pathogen's arsenal are known as "effectors;" these are secreted proteins that enter host cells to alter their physiology and suppress the immune system. Effectors can also be a liability for pathogens because potential hosts have evolved ways to recognize these effectors, triggering a defense response leading to localized cell death, preventing the spread of the pathogen. Here we computationally predicted effectors encoded in the genome of *Bremia lactucae*, the pathogen causing the downy mildew disease of lettuce and tested 21 of these proteins in the lab to see what effects they would have on the host plant. Five effectors were demonstrated to suppress the basal immune system of lettuce. Eleven effectors caused programmed cell death when introduced to certain lettuce lines, which indicates recognition of these proteins by the host immune system. In addition to contributing to our understanding of the mechanisms of pathogenesis, this study of effectors facilitates breeding for disease resistant lettuce, which will decrease agricultural reliance on fungicides.

## Introduction

Oomycetes are some of the most devastating pathogens of both plants and animals [1]. Although oomycetes resemble fungi in their filamentous growth and infection structures, they are more closely related to brown algae than to fungi [2]. Notable oomycetes include the plant pathogens causing late blight of potato (*Phytophthora infestans*) [3], sudden oak death (*Phytophthora ramorum*) [4], root rot (*Phytophthora* [5] and *Pythium* spp. [6,7]), white blister rust of *Brassica* spp. (*Albugo* spp.) [8], and downy mildews (e.g., *Bremia*, *Peronospora*, *Plasmopara* spp.) [9], as well as several important animal pathogens infecting fish (*Saprolegnia* spp.), shellfish (*Aphanomyces astaci*), and humans and other mammals (*Pythium insidiosum*) [1,10]. Many types of plant and animal pathogens, including oomycetes, secrete proteins known as effectors to promote virulence by manipulating the physiology of the host cells and by suppressing the host immune system [11]. In plants, effectors are also determinants of resistance or susceptibility through their direct or indirect interactions with cognate nucleotide-binding leucine rich repeat proteins (NLRs) encoded by resistance genes [12]. Effectors are secreted from the pathogen and may act extracellularly or they may be translocated into the cytoplasm [11]. One class of translocated effectors from plant pathogenic oomycetes of the class Peronosporales, which includes *Phytophthora* and the downy mildews, are the RXLR effectors, so called for their N-terminal motif usually consisting of arginine (R), followed by any amino acid (X), then followed by leucine (L) and arginine (R) [13]. RXLR effectors also contain an N-terminal signal peptide, which designates them for extracellular transport by way of the endoplasmic reticulum and Golgi apparatus [14]. The RXLR motif is often associated with a downstream EER motif, both of which have been associated with secretion and/or translocation of effectors into the plant cell [14,15]. For some RXLR effectors, such as Avr3a, the RXLR motif has been shown to be cleaved just prior to the EER sequence and therefore plays a role in targeting the protein for translocation rather than directly being involved in uptake into the host cell [16]. The RXLR motif is similar in sequence to the PEXEL motif (RXLX[EDQ]) of the distantly related malaria pathogen (*Plasmodium falciparum*) [17] and the TEXEL motif of *Toxoplasma gondii* (RRLXX) [18], both of which are required for proteolytic modification in the endoplasmic reticulum and targeting effector proteins for specialized export out of the cell [19].

The downy mildews and the related *Phytophthora* species have different lifestyles, with the downy mildews requiring a living host to complete their lifecycle (obligate biotrophy) and *Phytophthora* spp. using an initial biotrophic phase followed by a necrotrophic phase (hemibiotrophy) [20]; however, their effectors share similar features. Many effectors in the Peronosporales have RXLR and EER motifs, although several alternative sequences to RXLR have been found in downy mildews including RVRN (ATR5 from *Hyaloperonospora arabidopsidis*) [21], QXLR (*Pseudoperonospora cubensis*) [22], GKLR (*Bremia lactucae*) [23,24], and RXLK (*Plasmopara halstedii*) [25]. The C-terminal effector domains of RXLR effectors from *Phytophthora* and downy mildews also share some common sequence motifs and structural features, such as the 24 to 30 amino acid W, Y, and L motifs, which were first identified bioinformatically [26]. Structural analysis on four different RXLR effectors from *Phytophthora infestans* (Avr3a and PexRD2), *P. capsici* (Avr3a11), and the downy mildew pathogen *H. arabidopsidis* (ATR1) revealed that the W and Y motifs form a ~50 amino acid alpha-helical fold that may play a role in protein–protein interactions [27–30]. This effector-associated fold, termed the WY domain after its conserved tryptophan and tyrosine residues, is highly structurally conserved between effectors from multiple Peronosporales species, despite sharing less than 20% sequence similarity across the whole domain [27,31]. The WY domain appears to be specific to the Peronosporales and was predicted to be present in nearly half of the RXLR effectors of *P. infestans* and a quarter of the RXLR effectors in *H. arabidopsidis* [27].

Functional studies of WY domain containing proteins have indicated that certain residues in the WY domain are essential for the immune suppressing functions of *P. sojae* effectors Avr1b [32] and PSR1 [33], and *P. infestans* effectors Avr3a [34] and PexRD2 [35]. Furthermore, mutation of two conserved leucines in the WY domain of PexRD2 disrupted interaction with its target, MAPKKKε, consistent with this domain being important for protein–protein interactions [35]. WY domain containing proteins appear to interact with a variety of host targets, with Avr3a from *P. infestans* targeting the E3 ligase CMPG1 [34], *P. infestans* PexRD54 targeting potato autophagy-related protein ATG8 [36], PsAvh240 from *P. sojae* targeting an aspartic protease [37], and PSR1 and PSR2 from *P. sojae* suppressing host RNA silencing through interactions with the RNA helicase PINP1 and the dsRNA-binding protein DRB4, respectively [38–41]. Mutation analysis of the regions encoding the seven individual WY domains of PSR2 demonstrated differential contributions of each domain to virulence of *P. sojae*, suggesting that the WY domain may act as a module during effector evolution [31]. In addition to its roles in immune suppression, the WY domain has been shown to be important for immune recognition of the effector by nucleotide binding-leucine rich repeat (NB-LRR) resistance proteins [42].

Effector annotation in oomycete genomes has often relied on sequence similarity to known effectors or on prediction of conserved motifs, such as the RXLR motif, or in the case of Crinklers, the LXLFLAK motif [3]. Due to the short length and degeneracy of the RXLR sequence, the motif occurs frequently by chance; therefore, there is a high false positive rate (>50%) using string-based searches [26]. Hidden Markov Model (HMM)-based searches have much lower false positive rates, but the false negative rate may be higher if the genome of interest has diverged significantly from the species used to build the HMM. Downy mildews have a narrow host range and pathogenicity-related proteins are likely to show high lineage-specificity due to co-evolution with their hosts. There is already evidence that downy mildew effectors show divergence from the canonical RXLR motif [21,24,25]; thus, complementary approaches for effector prediction that utilize other conserved features, such as the WY domain, are necessary to fully characterize the repertoire. In addition, a WY domain containing protein from *Plasmopara viticola* lacking the RXLR motif has been found to trigger cell death in grapevine and tobacco species, suggesting that non-RXLR WY proteins may be recognized by resistance

proteins [43]. In addition, due to the importance of the WY domain in effector function [32,34,35], WY domain containing proteins may be better candidates for effectors than those containing only the RXLR motif.

To identify a more complete repertoire of candidate effectors in the reference genome of *B. lactucae* and to test whether the WY domain is informative for predicting effectors in the Peronosporales, we searched for this domain using an HMM built from sequences of the WY domain in three *Phytophthora* species [28]. This revealed additional effector candidates that had not been found using an RXLR-based search; similar results were also obtained for other downy mildews and well-studied *Phytophthora* species. These predicted WY proteins from *B. lactucae* had other signatures of oomycete effectors, such as presence of a secretion signal, N-terminal intrinsic disorder, lineage specificity, and expression during infection. A subset of the predicted WY effectors suppressed the host immune system, while others elicited programmed cell death in specific genotypes of lettuce, indicative of recognition by host resistance proteins. Therefore, searches for the WY domains are highly useful for identification of downy mildew effectors lacking the RXLR motif, which was previously considered canonical for effectors of the Peronosporales.

## Results

### Hidden Markov Model-based prediction of the WY domain identifies candidate effectors lacking the RXLR motif in *B. lactucae* and other oomycetes

Our HMM search initially identified 59 candidate WY effectors encoded by the gene models predicted in the *B. lactucae* SF5 assembly [44]; however, four pairs of genes appeared to be allelic based on sequence similarity and read depth, leaving a total of 55 non-redundant genes encoding candidate effectors (Fig 1A; S1 File). N-terminal signal peptides were predicted for 41 of these 55 proteins (Fig 1A), of which two proteins had a predicted transmembrane helix outside of the signal peptide. Several predicted WY containing proteins seemed to be missing their signal peptides due to N-terminal truncation when compared to close relatives. One of these predicted proteins, BLN06, was found to be missing 72 amino acids from the N-terminal sequence, including the signal peptide, in SF5 compared to BL24, which was the source isolate for the cloning of BLN06 reported by Pelgrom *et al.* [45]. BSW04p*, which appears to be allelic to BSW04p based on read depth, also has an N-terminal truncation resulting in loss of the signal peptide. All candidate WY effectors, including those without a signal peptide, are expressed during the lifecycle of *B. lactucae* as evidenced by RNAseq reads (S1 File) with expression confirmed by qRT-PCR for the 21 cloned genes (S1 Fig).

The N-terminal sequences of the 55 predicted WY effectors were examined for the RXLR motif using both HMMs and string searches. One protein was predicted to have an RXLR motif by the HMM and an additional 10 proteins were identified by a string search for [RQGH]XLR, while 33 proteins were predicted to have an EER motif using a string search for [DE][DE][RK] (Fig 1A). To find divergent RXLR-like motifs in the WY effector candidates, we searched for a highly degenerate pattern based on mutational studies of the RXLR motif [35] and natural RXLR variants reported for other downy mildews [5–9]. This revealed an additional 40 proteins with an RXLR-like motif within the first 60 amino acids after the signal peptide matching the pattern [RKHGQ][X]{0,1}[LMYFIV][RNK], many of which also had an EER motif (Fig 1A) (S1 Table). Therefore, the majority of candidate WY effectors in *B. lactucae* have a non-canonical RXLR motif.

In the C-terminal domain, the WY effector candidates had 1–8 WY domains per protein, with a diversity of domain architectures (Fig 1). The WY domain from *B. lactucae* showed high conservation of the characteristic conserved tryptophan (W) residue but appeared to

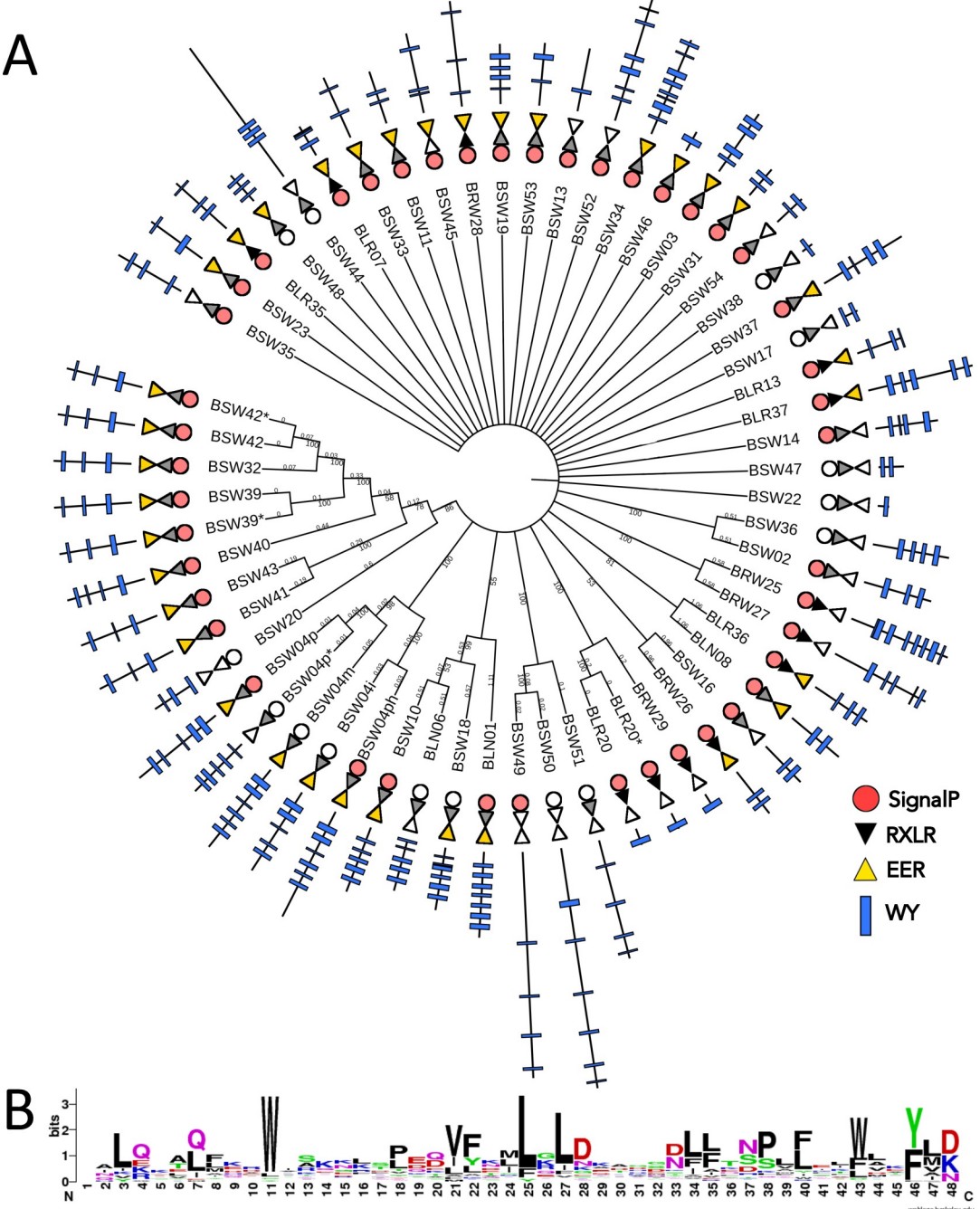

**Fig 1. WY effector candidates from *B. lactucae* isolate SF5.** (A) UPGMA consensus tree of the 59 predicted WY effectors from *B. lactucae* isolate SF5 based on whole protein amino acid sequence alignment using MUSCLE. Four sequences that appear to be allelic are indicated with asterisks next to the sequence name. Branch lengths have been transformed for clarity of presentation; however, bootstrap values (n = 100 iterations) and branch lengths are given for closely related proteins. Signal peptides are shown as circles (red circle for SP; white circle for no SP). RXLR motifs are shown as triangles (black triangle for RXLR ([RQGH]XLR or RXL[QKG]); grey triangle for degenerate RXLR ([RKHGQ][X]{0,1}[LMYFIV][RNK]), white triangle for no RXLR-like sequence) followed by an inverted triangle for EER motifs (yellow triangle for [DE][DE][RK], white triangle for no EER-like sequence). WY domain architecture is shown using blue rectangles, with a black line representing the total length of the protein and rectangle position representing the location of the WY motif as predicted by HMMer 3.1. (B) Sequence logo for the WY domain from *B. lactucae* built from a multiple sequence alignment of WY domains predicted by HMMer 3.1.

show equal preference for tyrosine (Y) or phenylalanine (F) for the second characteristic residue of the domain (Fig 1B).

## Many candidate WY effectors across oomycetes lack the RXLR motif

To investigate whether or not there are WY effector candidates that are lacking the canonical RXLR motif in other oomycetes, the predicted open reading frames (ORFs) from several published oomycete genomes were surveyed for the RXLR and EER motifs and the WY domain. Approximately half of the WY proteins predicted in seven downy mildew species lacked the RXLR motif; in six *Phytophthora* spp., the majority of predicted WY proteins had RXLR motifs, but between 9–21% of secreted WY proteins did not contain the RXLR motif (Fig 2; S2 Table). The majority of WY proteins lacking the RXLR motif contained the EER motif (Fig 2; S2 Table). Sequences of RXLR, EER, and WY-containing predicted proteins from all oomycete species analyzed can be found in S2 File. This analysis suggests that the repertoire of candidate WY effectors in other oomycetes may be heavily under-reported.

## Candidate WY effectors have high levels of N-terminal intrinsic disorder

Intrinsic disorder had previously been reported to be a characteristic of the N-terminus of oomycete effectors containing the RXLR motif [46]. Therefore, we investigated whether the predicted degree of structural disorder was a characteristic of candidate WY effectors lacking a

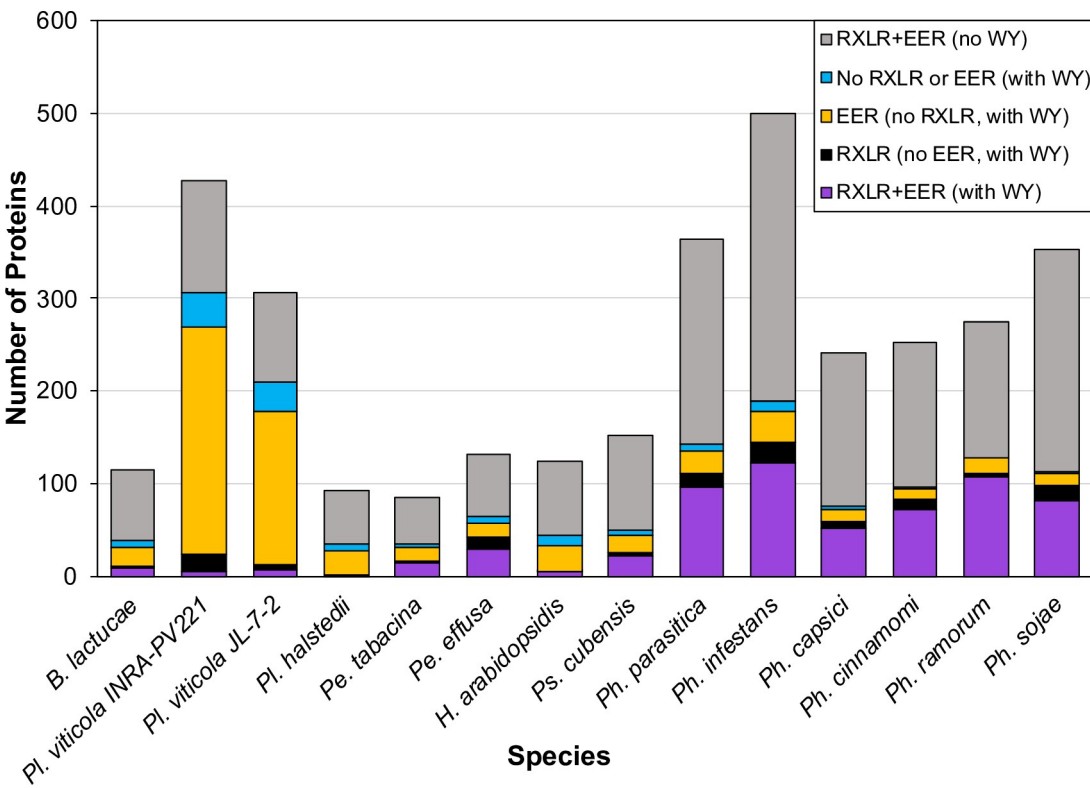

**Fig 2. Distribution of predicted secreted WY effectors with or without RXLR and/or EER motifs in the genomes of downy mildew pathogens and *Phytophthora* species.** HMMer was used to search predicted secretomes for each species for the WY domain. For the WY domaining containing proteins, the presence of RXLR and EER were determined by searching for [RQGH] XLR within the first 60 amino acids after the signal peptide and [DE][DE][KR] within the first 100 amino acids after the signal peptide. For the non-WY domain containing proteins, the RXLR and EER was determined by searching for a strict "RXLR" with [DE][DE][KR], within the first 60 and 100 amino acids, respectively, plus proteins found by searching for the RXLR-EER domain using the RXLR-EER HMM from [15].

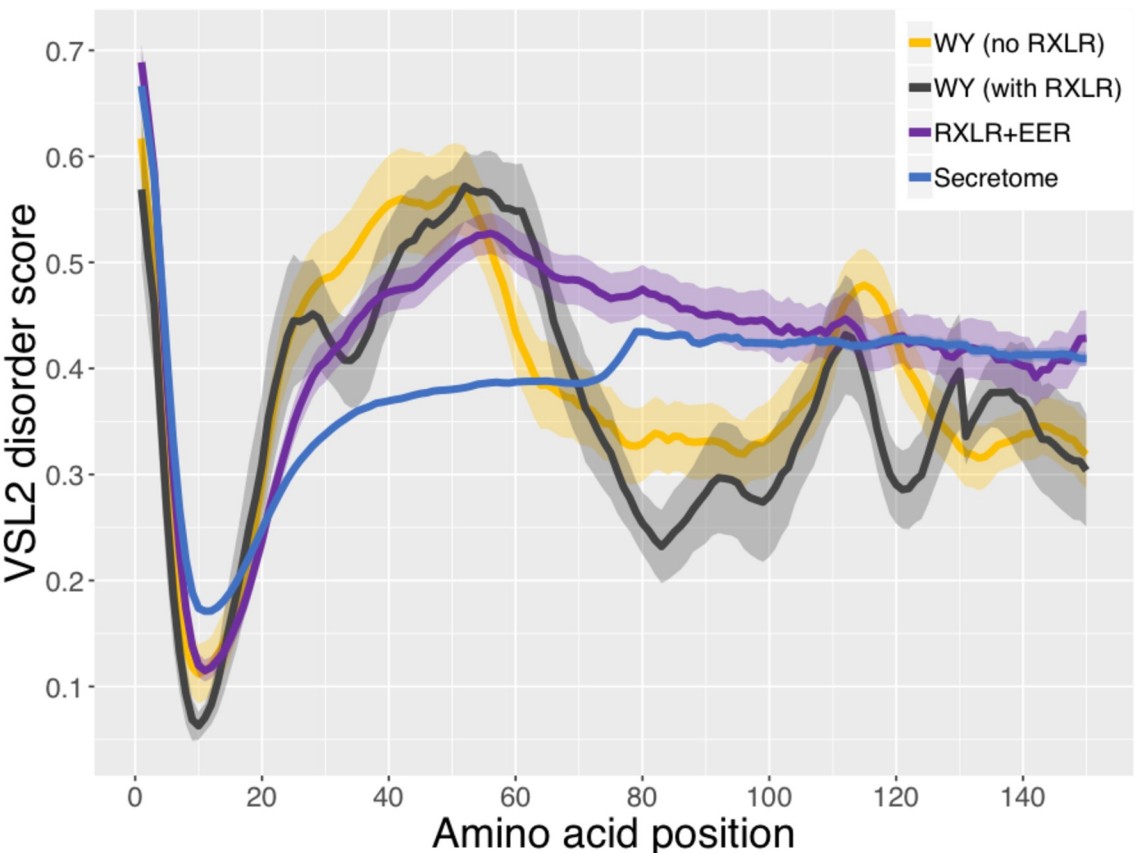

**Fig 3. Intrinsic disorder in the first 150 N-terminal amino acid sequences of RXLR and/or WY containing candidate effectors in *B. lactucae*.** Proteins were categorized as WY with no RXLR (yellow), WYs with RXLR (black), RXLR+EER (with or without WY, purple), and the total predicted secretome (blue). RXLR and EER motifs were as described in Fig 2. Average positional disorder scores [46] with standard error are shown for each group of proteins.

canonical RXLR. The predicted levels of intrinsic disorder were calculated for proteins containing RXLR motifs, for proteins containing WY domains but no RXLR motif, as well as for the entire predicted secretome for comparison. Proteins containing RXLR and/or WY domains had higher levels of intrinsic disorder at their N-termini after the highly ordered signal peptide than the entire set of secreted proteins (Fig 3). Proteins containing a WY domain but lacking a canonical RXLR motif had on average more disordered N-termini than effectors that had RXLR but not a WY domain (Fig 3). The WY domain containing region had higher levels of predicted structure than the RXLR-containing proteins that lacked a WY domain and the secreted proteins as a whole (Fig 3), consistent with the WY domains forming an α-helical bundled structure [28]. This predicted pattern of high intrinsic disorder at the N-terminus and high structure towards the C-terminus was consistently observed in all six downy mildews and six *Phytophthora* species analyzed (S2 Fig). Therefore, a high level of intrinsic disorder is a consistent characteristic of the N-termini of oomycete effectors, regardless of whether they have a canonical RXLR motif; the functional significance of this remains to be investigated.

## Candidate WY effectors show lineage specificity to *B. lactucae*

To evaluate lineage specificity of effectors due to co-evolution of pathogens with their hosts, we used BLAST to identify orthologs in other oomycete species. All of the 41 predicted

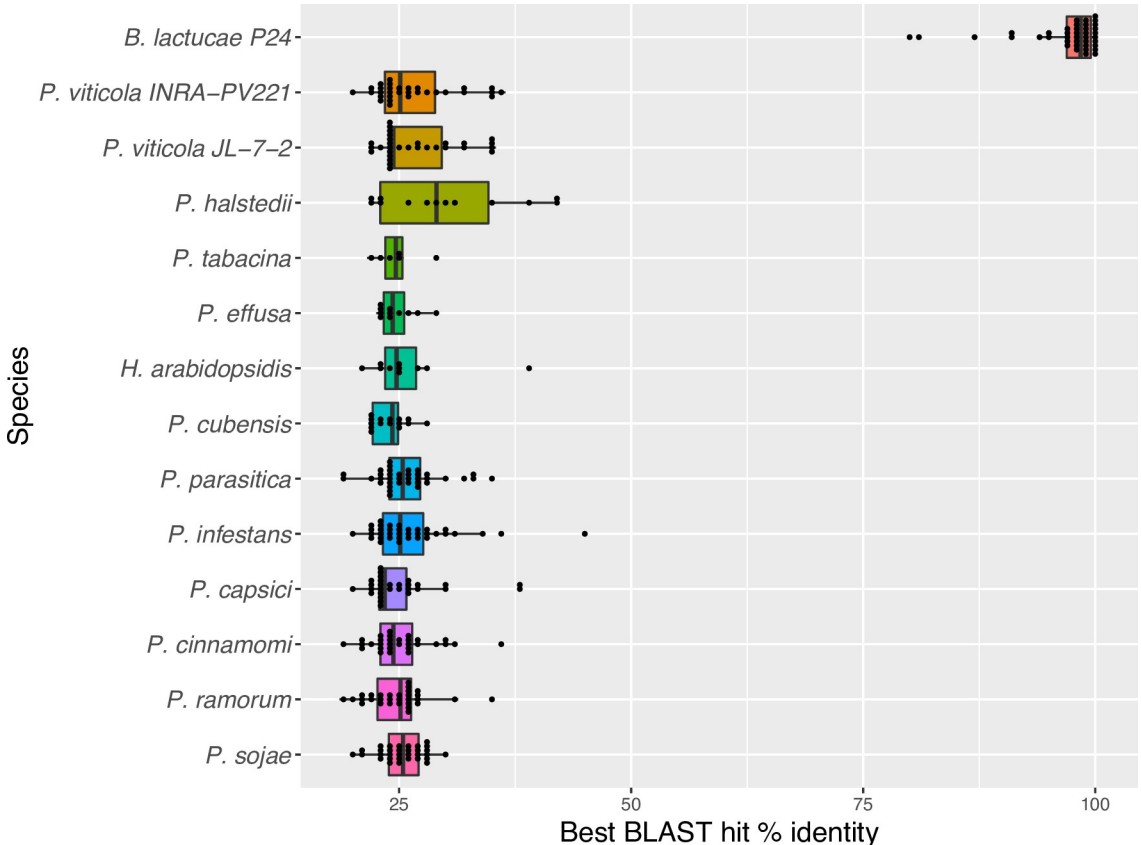

**Fig 4. Lineage specificity of predicted secreted WY effectors from *B. lactucae*.** Secreted WY proteins predicted in isolate SF5 of *B. lactucae* were used as a query for BLASTp against other oomycete translated ORFs. Isolate C82P24 was queried for *B. lactucae*. The best BLAST hit percent identity for each protein was calculated. The box plot shows the distribution of the best BLAST hits per *B. lactucae* WY protein from each species, with each dot representing an individual data point. No hits were observed to *Albugo laibachii*, *Saprolegnia parasitica*, or *Pythium ultimum*.

secreted candidate WY effectors of *B. lactucae* had little sequence similarity to sequences in other genomes with the best BLASTp hit having only 46% identity with a protein from *P. infestans* (Fig 4). Most of the proteins had best-hit identities between 20 to 30%, which is similar to the level of amino acid conservation between WY domains in different effectors [28]. All of the proteins had hits to *B. lactucae* isolate C82P24, with very high identity, indicating conservation of these candidate effectors in *B. lactucae* (Fig 4).

## Candidate WY effectors show distinct subcellular localizations when transiently expressed in lettuce

Eight *B. lactucae* WY effectors were tagged at their N-terminus with either yellow fluorescent protein (YFP) or mCherry and their subcellular localizations were visualized using confocal microscopy (Fig 5). Five of the WY effectors (BLN08, BSW03, BLN06, BSW11, and BSW13) were localized to the cytoplasm and/or plasma membrane and excluded from the nucleus. BSW04p was localized to the nucleus, consistent with its C-terminal nuclear localization signal (NucPred [47] score 0.79). BSW19 showed both nuclear and cytoplasmic localization and also showed localization to round bodies in the cell. BSW14 was excluded from the nucleus and appeared to be localized to round bodies. Full length tagged proteins were observed for all eight tagged WY effectors by Western blot (S3 Fig). Some effectors showed cleavage of

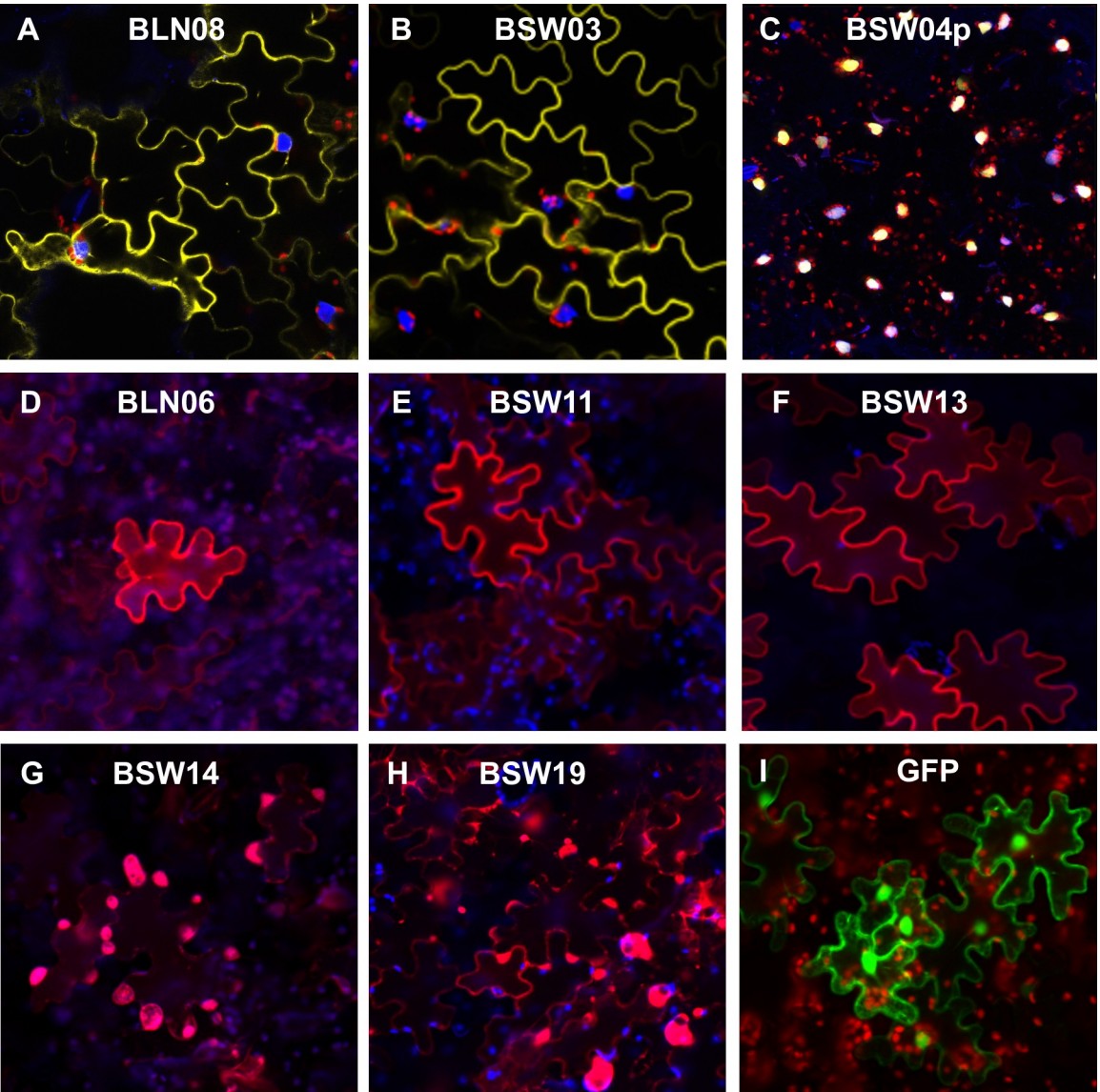

**Fig 5. Confocal microscopy of fluorescently-tagged WY containing candidate effectors from *B. lactucae* transiently expressed in lettuce.** Panels A-C show N-terminal YFP-tagged effectors (yellow), DAPI nuclear stain (blue), and chlorophyll autofluorescence (red). Panels D-H show N-terminal mCherry-tagged effectors (red) and chlorophyll autofluorescence (blue). Panel I shows free GFP (green) and chlorophyll autofluorescence (red). All squares are 200 x 200 μM.

mCherry or YFP on the Western blot (S3 Fig); however, since their localization patterns are distinct from free GFP (Fig 5), this degradation likely occurred during protein extraction.

### Several candidate WY effectors suppress PTI-induced ROS activity

We tested candidate WY effectors for their ability to suppress PAMP-triggered immunity (PTI). The cloned candidate WY effectors were transiently expressed in *Nicotiana benthamiana* and the level of reactive oxygen species (ROS) production induced by flg22 was measured. Several effectors significantly suppressed ROS production to a similar extent as two known bacterial suppressors of PTI (Fig 6). Interestingly, the addition of N-terminal fluorescent tags on some of the effectors influenced their ability to suppress PTI. For BSW04p and BSW13, the

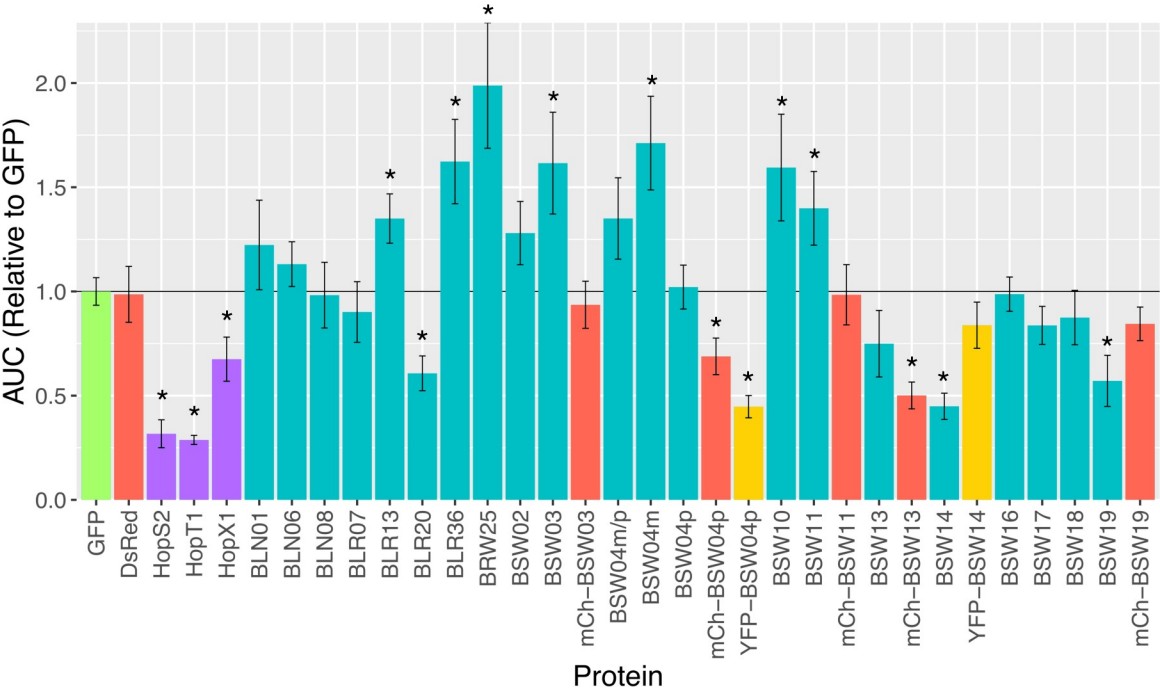

**Fig 6. Pattern-triggered immunity suppression ability of candidate WY containing effectors from *B. lactucae* as measured by flg22-triggered reactive oxygen species burst.** Bar plot showing averages and standard error in PTI response, measured by the area under the curve (AUC) of luminescence during the 40-minute ROS assay, for leaf discs expressing each effector relative to leaf discs expressing the GFP control. DsRed was used as an additional negative control. *P. syringae* Hop effectors were used as positive controls for PTI suppression (purple bars). Tagged effectors are indicated by mCh (mCherry, red bars) or YFP (yellow fluorescent protein, yellow bars). Asterisks indicate PTI responses that are statistically different from the GFP control (t-test, $p < 0.05$).

addition of YFP or mCherry tags increased the suppression ability, possibly by providing higher expression or increased protein stability; for BSW14 and BSW19, the addition of YFP and mCherry tags reduced the ability of these effectors to suppress PTI (Fig 6). All fluorescently-tagged proteins were confirmed to be full length by Western blotting, although all effectors, except for BSW04p, also showed a band representing cleaved mCherry or YFP (S4 Fig). The level of induction of ROS by flg22 was significantly higher with some effectors; however, there was no induction of ROS in the absence of flg22. Therefore, the results from this assay suggest that at least five effectors can suppress PTI and some may actually increase PTI responsiveness.

## Multiple candidate WY effectors trigger cell death in certain lettuce genotypes

The 21 cloned, untagged WY candidate effectors were screened for their ability to elicit cell death in 215 different accessions of wild and cultivated lettuce (S3 File) using *Agrobacterium*-mediated transient expression. These accessions collectively express the majority of the known *Dm* genes as well as new resistance factors [48]. Eleven of the 21 WY effectors (BLR13, BLR36, BLN06, BLN08, BSW03, BSW04m, BSW04p, BSW11, BSW13, BSW14, and BSW19) triggered chlorosis or necrosis in one or more accessions of lettuce (Fig 7). Alleles of the same effector showed similar patterns of necrosis and/or chlorosis except for BSW11 by UCDM14 and PI491178 (Fig 7). For BLN06, we found that the allele cloned from SF5, which lacks a signal peptide, caused necrosis in NunDm17 and chlorosis in RYZ2146 (S5 Fig), as was previously

| Genotype | species | BLR13 | BLR36 | BLN06 | BLN08 | BSW03 | BSW04m | BSW04m/p | BSW04p | BSW11a | BSW11b | BSW13 | BSW14 | BSW19 | GFP | +control |
|---|---|---|---|---|---|---|---|---|---|---|---|---|---|---|---|---|
| CGN14312 | indica | 1.3 | 3.0 | 3.3 | - | - | - | - | - | 1.5 | 1.5 | 1.5 | 2.8 | 1.0 | 0.8 | 3.9 |
| CGN14316 | indica | 0.8 | 1.5 | 0.0 | 0.5 | 0.4 | 1.5 | 2.3 | 1.0 | 1.0 | 1.8 | 1.6 | 0.6 | 0.0 | 0.4 | 3.8 |
| 25 acc. | saligna | 0.0 | 0.4 | 0.0 | 3.2 | 0.2 | 0.2 | 0.3 | 0.2 | 0.0 | 0.0 | 0.0 | 3.6 | 0.1 | 0.1 | 3.9 |
| 11 acc. | saligna | 0.0 | 2.9 | 0.0 | 3.6 | 0.4 | 0.3 | 0.5 | 0.3 | 0.0 | 0.0 | 0.1 | 4.0 | 0.0 | 0.2 | 4.0 |
| CGN10888 | saligna | 0.0 | 0.0 | 0.0 | 0.8 | 0.0 | 0.0 | 0.0 | 1.0 | 0.0 | 0.0 | 0.0 | 4.0 | 0.0 | 0.0 | 4.0 |
| CGN11341 | saligna | 0.0 | 0.0 | 0.0 | 1.6 | 0.5 | 0.3 | - | 0.4 | 0.0 | 0.0 | 0.0 | 3.8 | 0.0 | 0.0 | 3.8 |
| CGN13326 | saligna | 0.0 | 1.7 | 0.0 | 3.7 | 0.0 | 0.3 | 0.5 | 0.0 | 0.0 | 0.0 | 0.0 | 4.0 | 0.0 | 0.2 | 4.0 |
| CGN13327 | saligna | 0.0 | 0.0 | 0.0 | 0.0 | 0.0 | 0.0 | 0.0 | 0.0 | 0.0 | 0.0 | 0.0 | 4.0 | 0.0 | 0.0 | 4.0 |
| CGN13330 | saligna | 0.0 | 0.3 | 0.3 | 1.3 | 0.5 | 0.6 | 1.0 | 0.3 | 0.2 | 0.0 | 0.4 | 4.0 | 0.0 | 0.3 | 4.0 |
| CGN15716 | saligna | 0.0 | 0.0 | 0.0 | - | - | - | - | - | 0.0 | 0.0 | 0.0 | 4.0 | 0.0 | 0.0 | 4.0 |
| CGN5282 | saligna | 0.0 | 0.0 | 0.0 | 2.5 | 0.0 | 0.0 | 1.5 | 0.0 | 0.0 | 0.0 | 4.0 | 4.0 | 0.0 | 0.0 | 4.0 |
| CGN5309 | saligna | 0.0 | 0.8 | 2.5 | 3.7 | 0.3 | 0.6 | 0.3 | 1.0 | 0.0 | 0.0 | 4.0 | 4.0 | 0.0 | 0.3 | 4.0 |
| CGN5314 | saligna | 0.0 | 0.8 | 3.3 | 3.8 | 0.8 | 0.8 | 0.8 | 1.5 | 0.0 | 0.0 | 0.0 | 4.0 | 0.0 | 0.4 | 4.0 |
| CGN5315 | saligna | 0.0 | 0.8 | 1.8 | 3.9 | 0.3 | 1.0 | 0.8 | 1.9 | 0.0 | 0.0 | 0.0 | 4.0 | 0.0 | 0.0 | 4.0 |
| CGN5318 | saligna | 0.0 | 1.3 | 0.0 | 3.5 | 0.8 | 0.3 | 2.0 | 0.3 | 0.0 | 0.0 | 0.0 | 4.0 | 0.0 | 0.1 | 3.7 |
| CGN5322 | saligna | 0.0 | 0.0 | 0.0 | 3.5 | 0.2 | 0.1 | 1.0 | 2.4 | 0.0 | 0.0 | 0.0 | 4.0 | 0.0 | 0.1 | 4.0 |
| CGN5329 | saligna | 0.0 | 0.0 | 0.0 | 2.4 | 0.1 | 0.9 | 1.0 | 1.8 | 0.0 | 0.0 | 0.0 | 2.8 | 2.5 | 0.3 | 3.9 |
| CGN5330 | saligna | 0.0 | 0.0 | 0.0 | 3.3 | 0.0 | 0.6 | 1.0 | 3.1 | 0.0 | 0.0 | 0.0 | 3.5 | 0.0 | 0.1 | 4.0 |
| LB108 | saligna | 0.0 | 0.0 | 0.0 | 1.8 | 0.0 | 0.0 | 0.0 | 0.0 | 0.0 | - | 0.0 | 2.0 | 0.0 | 0.0 | 4.0 |
| PI491000 | saligna | 0.0 | 1.0 | 0.0 | 4.0 | 0.0 | 0.0 | 0.0 | 0.0 | 0.3 | 0.0 | 1.5 | 1.6 | 0.0 | 0.0 | 3.9 |
| PI491207 | saligna | 0.0 | 0.0 | 0.0 | 4.0 | 1.1 | 2.0 | 2.0 | 0.5 | 0.0 | 0.0 | 0.0 | 4.0 | 0.0 | 0.2 | 4.0 |
| PI491208 | saligna | 0.0 | 0.0 | 0.0 | 3.6 | 0.6 | 0.6 | 2.0 | 3.6 | 0.0 | 0.0 | 0.0 | 4.0 | 0.0 | 0.7 | 4.0 |
| Blonde Lente | sativa | 0.5 | 0.5 | 0.0 | 0.0 | 0.5 | 0.5 | 1.0 | 0.0 | 2.0 | 2.7 | 0.5 | 0.0 | 0.0 | 0.2 | 4.0 |
| Capitan | sativa | 0.0 | 0.0 | 0.0 | 0.6 | 0.2 | 0.2 | 3.3 | 3.5 | 0.0 | 0.0 | 0.0 | 0.0 | 0.0 | 0.2 | 3.8 |
| Capsule | sativa | 0.0 | 0.0 | 0.0 | - | - | - | - | - | 0.0 | 0.0 | 0.0 | 4.0 | 0.0 | 0.0 | 4.0 |
| Femke | sativa | 0.0 | 0.0 | 0.0 | 0.0 | 0.0 | 2.4 | 3.8 | 3.0 | 0.0 | 0.0 | 0.0 | 0.0 | 0.0 | 0.0 | 4.0 |
| Fenston | sativa | 0.0 | 0.0 | 1.0 | 0.0 | 0.6 | 1.3 | 1.5 | 2.8 | 0.0 | 0.0 | 0.0 | 0.0 | 0.0 | 0.4 | 3.6 |
| FrRsal-1 | sativa | 0.0 | 1.5 | 0.0 | 0.0 | 0.0 | 0.1 | 3.4 | 0.0 | 0.0 | 0.0 | 0.0 | 0.0 | 0.0 | 0.0 | 3.8 |
| Ninja | sativa | 0.0 | 1.2 | 0.0 | 0.0 | 0.0 | 0.2 | 2.0 | 3.4 | 0.0 | 0.6 | 0.0 | 0.0 | 0.0 | 0.2 | 4.0 |
| Pennlake | sativa | 0.0 | 0.0 | 0.0 | 0.0 | 0.0 | 0.0 | 2.1 | 0.0 | 0.4 | 0.0 | 0.0 | 0.0 | 0.0 | 0.0 | 3.5 |
| PI491226 | sativa | 0.0 | 0.0 | 0.0 | 3.7 | 0.6 | 0.5 | 1.0 | 1.5 | 0.0 | 0.0 | 0.0 | 4.0 | 0.0 | 0.1 | 3.9 |
| RYZ2164 | sativa | 0.0 | 0.0 | 0.0 | 0.0 | 0.0 | 0.0 | 3.3 | 0.0 | 0.0 | 0.0 | 0.0 | 0.0 | 0.0 | 0.0 | 4.0 |
| Salvius | sativa | 0.0 | 0.0 | 0.0 | 0.0 | 0.0 | 0.0 | 2.0 | 0.0 | 0.0 | 0.0 | 0.0 | 0.0 | 0.0 | 0.0 | 3.6 |
| UCDM14 | sativa | 0.0 | 0.0 | 0.0 | 0.1 | 0.1 | 0.0 | 0.4 | 0.0 | 3.1 | 0.0 | 0.0 | 0.2 | 0.0 | 0.4 | 3.8 |
| Versaï | sativa | 0.0 | 0.0 | 0.4 | 0.0 | 0.0 | 0.7 | - | 0.0 | 3.2 | 2.3 | 0.0 | 0.0 | 0.0 | 0.0 | 4.0 |
| ViAE | sativa | 0.0 | 0.0 | 0.0 | 3.4 | 3.5 | 3.9 | 2.0 | 3.8 | 0.0 | 0.0 | 0.0 | 2.8 | 0.0 | 0.1 | 3.7 |
| ViCQ | sativa | 1.0 | 0.0 | 0.0 | 3.9 | 2.8 | 3.8 | 3.3 | 3.4 | 0.0 | 0.0 | 0.0 | 3.2 | 0.0 | 0.1 | 3.8 |
| Arm999 | serriola | 0.0 | 0.0 | 0.0 | 0.0 | 0.0 | 3.6 | 4.0 | 1.0 | 0.0 | 0.0 | 0.0 | 0.6 | 0.0 | 0.0 | 4.0 |
| CGN14280 | serriola | 2.2 | 0.0 | 0.5 | 0.3 | 0.0 | 0.0 | 0.3 | 0.4 | 0.0 | 0.0 | 0.0 | 0.0 | 0.0 | 0.3 | 3.3 |
| ISR-380 | serriola | 0.5 | 0.5 | 1.0 | 0.0 | 1.0 | 0.5 | 0.5 | 0.5 | 0.5 | - | 0.3 | 1.0 | 2.5 | 0.4 | 4.0 |
| LS102 | serriola | 0.0 | 0.0 | 0.0 | 0.0 | 0.3 | 0.0 | 0.0 | 0.0 | 0.0 | 0.0 | 1.0 | 2.3 | 1.3 | 0.2 | 4.0 |
| PI491108 | serriola | 0.0 | 0.0 | 0.0 | 2.1 | 0.0 | 0.0 | 0.0 | 0.0 | 0.0 | 0.0 | 0.0 | 3.3 | 0.0 | 0.0 | 4.0 |
| PI491178 | serriola | 0.0 | 0.0 | 0.0 | - | 0.0 | 0.0 | 0.0 | 0.0 | 4.0 | 0.0 | 0.0 | 0.0 | 0.0 | 0.0 | 4.0 |
| CGN14305 | virosa | 0.0 | 0.0 | 0.0 | 1.5 | 0.7 | 0.9 | 1.5 | 1.4 | 0.0 | 0.0 | 0.0 | 2.3 | 0.0 | 0.4 | 3.6 |
| CGN4683 | virosa | - | - | - | 3.5 | 0.0 | 0.0 | - | 0.0 | - | - | - | - | - | 0.0 | 4.0 |
| CGN5333 | virosa | 0.0 | 0.0 | 0.5 | 0.0 | 0.1 | 0.0 | 0.0 | 0.0 | 0.0 | 0.0 | 0.0 | 2.3 | 1.5 | 0.0 | 3.9 |
| CGN9365 | virosa | 0.0 | 0.0 | 0.0 | 0.2 | 0.0 | 0.8 | 0.0 | 0.0 | 0.0 | 0.0 | 0.0 | 3.8 | 0.0 | 0.0 | 4.0 |
| LS238 | virosa | 0.0 | 0.0 | - | 1.8 | 2.0 | 3.9 | 0.0 | 3.6 | 0.0 | 0.0 | 0.0 | 2.0 | 0.0 | 0.0 | 3.4 |
| LS241 | virosa | 0.0 | 0.0 | 0.0 | 2.5 | 2.3 | 3.9 | 0.0 | 3.2 | 0.0 | 0.0 | 0.0 | 3.5 | 0.0 | 0.0 | 3.1 |

| Color Key | Score |
|---|---|
| No reaction | 0 |
| Mild chlorosis | 1 |
| Chlorosis | 2 |
| Mild Necrosis | 3 |
| Necrosis | 4 |
| No data | - |

**Fig 7. Agroinfiltration results from the screen of diverse genotypes of lettuce for recognition of candidate WY containing effectors from *B. lactucae* indicated by necrosis and/or chlorosis.** Candidate effectors were expressed in lettuce leaves using Agroinfiltration and leaves were scored for their reaction four to seven days post-infiltration. Qualitative scores were converted into numeric scores for data analysis: 0 = no reaction (green), 1 = mild chlorosis (light green), 2 = chlorosis (yellow), 3 = mild necrosis or mixture of chlorosis and necrosis (pink), and 4 = full necrosis (magenta). The figure shows average scores for each effector on each genotype; only accessions and effectors that had a necrotic or chlorotic response are shown. GFP was used for a negative control and either HopM1 or PsojNIP was used as a positive control; only leaves that showed an appropriate response to controls were scored. The two effectors containing RXLR motifs are highlighted in dark grey, while effectors with no RXLR motif are highlighted in blue. Scores for all accessions screened and sample sizes can be found at http://bremia.ucdavis.edu/BIL/BIL_interaction.php. Two groups of 28 and 11 accessions of *L. saligna* that had identical reactions are shown together; the individual accessions making up each group are shown in S3 File.

found for full length BLN06 from the isolate BL24 [45]. BLN06 from SF5 did not cause necrosis or chlorosis on LS102 despite being expressed (S5 Fig). BLN06 from BL24 causes chlorosis in LS102 [45]; however, without simultaneous comparisons with a clone of BLN06 from BL24, the reasons for this difference are unclear.

Two effectors, BLN08 and BSW14, triggered cell death on many accessions of *L. saligna*. Cell death was observed for additional effectors, up to four per genotype, on several accessions of *L. saligna* (Fig 7; S2 File). Two lines of *L. sativa* tested, ViAE and ViCQ, which both have introgressions from the wild lettuce species *L. virosa* [49], were observed to have cell death in response to five effectors (BLN08, BSW03, BSW04m, BSW04p, and BSW14) as well as the chimeric sequence BSW04m/p (Fig 8). BSW04m/p was cloned using the same primers as BSW04p but did not align to a single gene in the reference genome and was therefore determined to be a chimera between the paralogous sequences BSW04p and BSW04m. The *L. virosa* accessions that were the resistance donors for ViAE and ViCQ also were observed to have cell death in response to these five effectors, but not the chimeric sequence BSW04m/p (Fig 8). To assess whether effector triggered cell death is genetically linked to resistance to *B. lactucae* in ViAE, we tested 67 $F_{8:9}$ recombinant inbred lines (RILs) of Cobham Green x ViAE [49] using agroinfiltration of the five effectors and by screening these lines for resistance to *B. lactucae*. There were 43 resistant RILs, 19 susceptible RILs, and five segregating RILs. All five effectors triggered cell death in all the resistant RILs and none triggered cell death in the susceptible RILs (Fig 9), demonstrating genetic linkage between effector triggered cell death and resistance to *B. lactucae*. Effector response was also found to be segregating in the five RILs that segregated for resistance to *B. lactucae*.

Many *L. sativa* lines and a few genotypes of *L. saligna* and *L. serriola* were observed to have necrosis or yellowing in response to the chimeric effector BSW04m/p. Some of the genotypes that had effector triggered cell death caused by BSW04m/p also had cell death in response to BSW04p, but not the paralog BSW04m, which shares 91% amino acid identity with BSW04p. Three lines in the original screen, Capitan, Ninja, and Femke, share the resistance gene *Dm11*; therefore, we tested two additional lines, Fila and Mondian, which also contain *Dm11*, for recognition of BSW04p. These lines similarly had effector triggered cell death caused by BSW04m/p and BSW04p, but not BSW04m. Since BSW04p causes cell death on multiple varieties that contain *Dm11*, BSW04p is a candidate for the protein encoded by the *Avr11* gene.

## A region in the C-terminus of BSW04p is responsible for cell death on *L. sativa* cv. Ninja

To determine the region of the BSW04p sequence responsible for cell death on Ninja, we generated four domain-swapped chimeric proteins between BSW04m (which does not cause cell death on Ninja, despite being expressed (S6 Fig)) and BSW04p (which causes cell death on Ninja) at different locations along the protein (Fig 10C). Only the chimeras with the C-

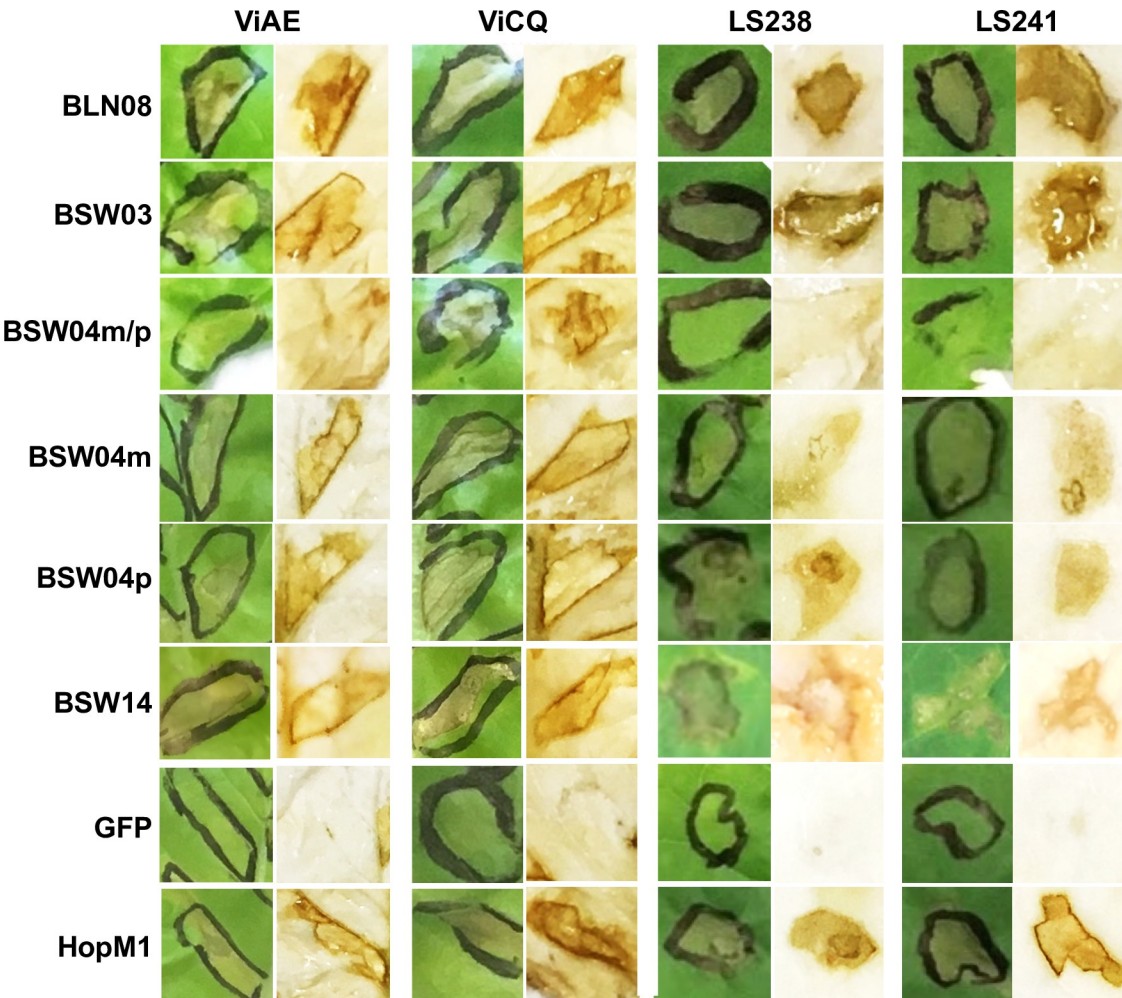

**Fig 8. Example agroinfiltration results of reactions of *L. sativa* ViAE and ViCQ and their respective progenitor R gene donors *L. virosa* LS241 and LS238 to candidate WY containing effectors from *B. lactucae*.** Photos are representative of a typical leaf. GFP and HopM1 were used as negative and positive controls for necrosis, respectively. Leaves were collected five days post-infiltration and the first column for each accession shows the uncleared leaf tissue; the second column shows leaf tissue cleared in ethanol to make necrosis (brown areas) more visible.

terminal domain of BSW04p elicited cell death on Ninja (Fig 10). Sequence comparisons of all the isoforms tested (both the domain swaps and alleles obtained through the original cloning experiment) narrowed down the causative region for cell death on Ninja between residues 405 and 458, which contains five amino acid polymorphisms between cell death triggering and non-cell death triggering sequences (Fig 10D). All of the other C-terminal polymorphisms outside of this region were found to be present in the unrecognized BSW04m alleles BSW04m-10C or BSW04m-4C. The causative region is located between two predicted WY domains (Fig 10C).

## Discussion

Effectors play a critical role in interactions between pathogens and their hosts. Accurate prediction and annotation of effector repertoires is foundational for functional genomics studies of pathogens. Due to their economic importance in agriculture, an increasing number of

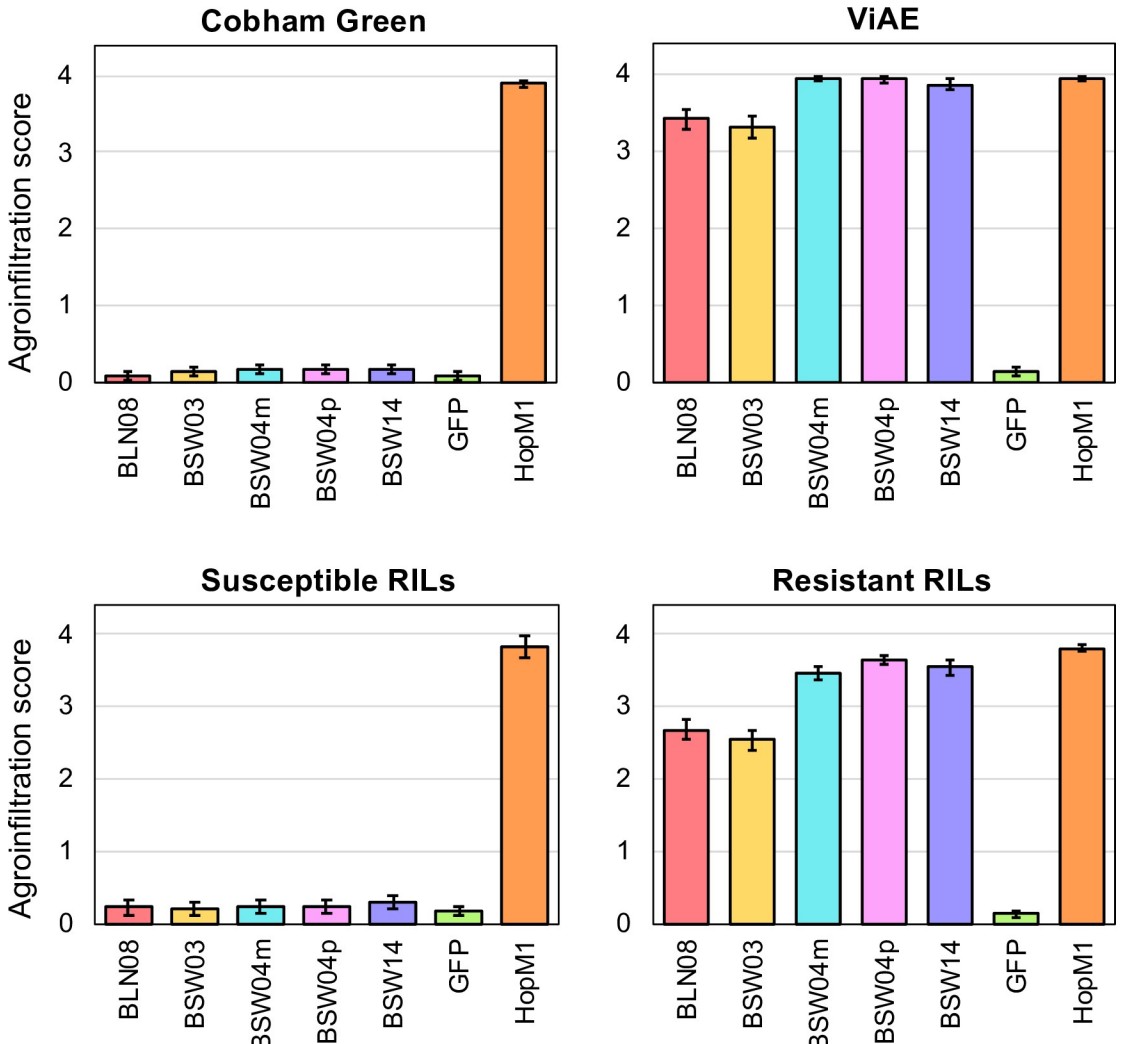

**Fig 9. Agroinfiltration scores of the ViAE x Cobham Green RILs in response to the five effectors recognized by ViAE.** Scores as in Fig 7. Candidate effectors were expressed in lettuce leaves using Agroinfiltration and leaves were scored for their reaction three to five days post-infiltration. GFP was used for a negative control and HopM1 used as a positive control for all leaves; only leaves showing the appropriate response to both were scored. Sample sizes for each group are 35 plants of Cobham Green, 45 plants of ViAE, 18 lines of susceptible RILs (88 plants), and 44 lines of resistant RILs (205 plants). Bar plot shows average scores with standard error bars.

*Phytophthora* and downy mildew pathogens are being sequenced. Prior to this study, the majority of cloned avirulence genes have encoded proteins with a RXLR motif [50]. This was consistent with most bioinformatic studies using the RXLR motif as their primary, or only, criterion to identify candidate effectors. Our results reveal that there are many candidate effectors containing the effector-related WY domain that lack the canonical RXLR motif, especially in the downy mildews, but also in *Phytophthora* spp. In *B. lactucae*, we have shown that these non-RXLR, WY domain containing effectors show characteristics of RXLR effectors, such as N-terminal intrinsic disorder, lineage specificity, and expression during infection. Furthermore, some of these proteins can act as suppressors of the host immune system and some trigger cell death in certain lettuce lines. Therefore, despite lacking the canonical RXLR-motif, WY domain containing proteins can have both virulence and avirulence activities. Consequently, numerous candidate effector genes are likely to have been missed in the genomic

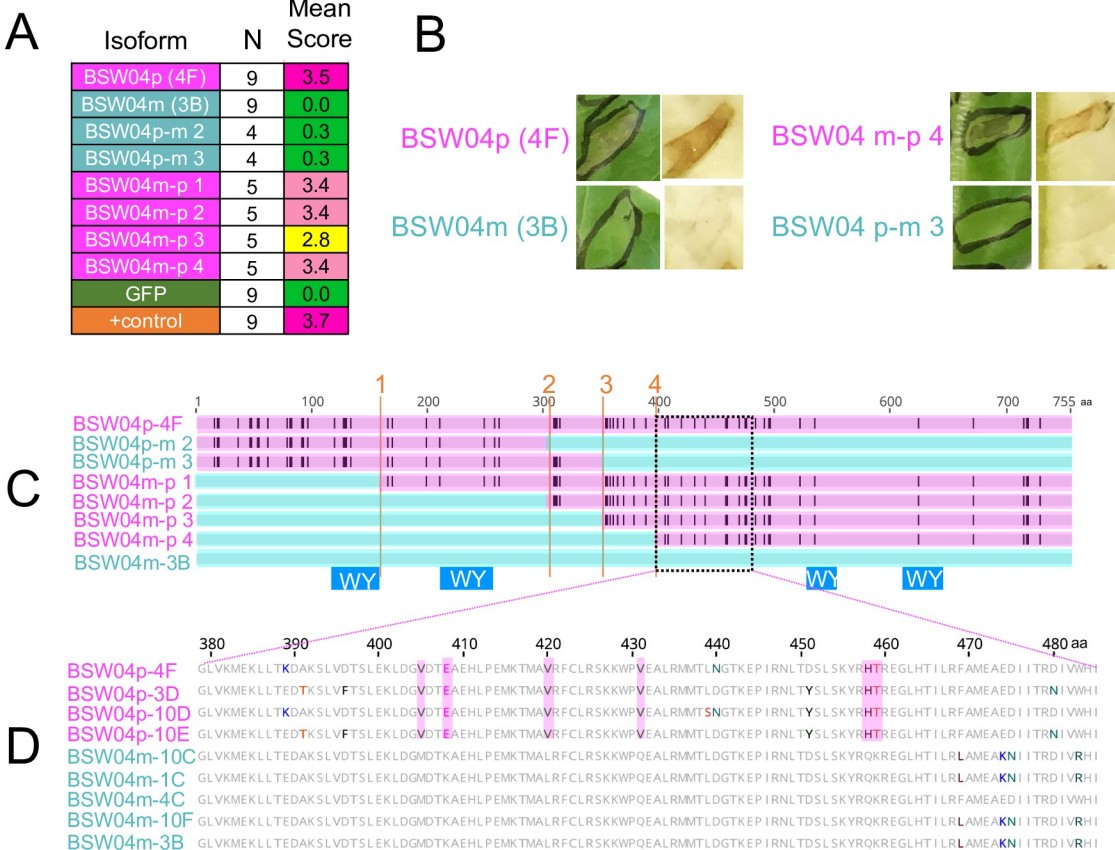

**Fig 10.** **(A) Agroinfiltration scores for the BSW04 paralogs and domain swaps.** One leaf per plant was infiltrated and scores were averaged across N replicates. Scoring and color scheme as in Fig 7. **(B) Example agroinfiltration results from BSW04m-BSW04p domain swaps.** Infiltrations were done in *L. sativa* cv. Ninja and scored 5 days post infiltration. **(C) Sequence alignment of BSW04m, BSW04p, and the domain swaps.** Black lines on sequence alignment indicate amino acid differences from BSW04m. Orange lines indicate recombination breakpoints used in generating the domain swaps. Blue squares show the predicted WY domains. Proteins with sequences that cause necrosis on Ninja are colored magenta; those that do not are colored aqua. **(D) Sequence alignment of the putative recognized region of BSW04p for recognized and unrecognized clones of BSW04.**

analysis of other species within the Peronosporales. Bioinformatics pipelines for predicting effectors in both *Phytophthora* spp. and downy mildew pathogens should include an HMM for the WY domain because it has a low false positive rate (Table 1) and may discover more high confidence effectors than searches for the RXLR motif alone.

Some WY effectors were initially predicted to not have a signal peptide and therefore not to be secreted; this could have been due to misannotation or may reflect biological reality. Several

**Table 1. False positive rates for HMM and string searches across 14 oomycete genomes.**

| Motif | String/HMM | False Positive Rate (Average +SD) | Range |
|---|---|---|---|
| RXLR | RXLR | 45.2% ± 3.4% | 32–54% |
| RXLR | [RQGH]XLR | 55.2% ± 3.1% | 44–61% |
| RXLR | Whisson HMM [15] | 1.5% ± 0.8% | 0.1–3.3% |
| EER | [DE][DE][KR] | 49.4% ± 2.6% | 41–62% |
| RXLR-EER | [RQGH]XLR + [DE][DE][KR] | 15.4% ± 4.7% | 7.0–23% |
| Degenerate RXLR | [RKHGQ][X]{0,1}[LMYFIV][RNK] | 74%± 0.8% | 71–77% |
| WY | Boutemy HMM [28] | 0.05% ± 0.09% | 0–0.3% |

appeared to be misannotated since they had a predicted signal peptide starting at the second start codon of the gene model. Therefore, studies that rely on the presence of a signal peptide when using predicted ORFs may be missing true effectors with misannotated start codons. Some WY effectors had clearly lost the signal peptide due to N-terminal deletion, such as BLN06 (relative to the isolate BL24 [45]) and BSW04m (relative to its paralog BSW04p). Hypothetically, signal peptide loss may be an evolutionary strategy for the pathogen to evade recognition of an effector while retaining the effector function encoded in the genome for future deployment.

Although only 10% of WY effectors from *B. lactucae* contained the canonical RXLR motif, nearly all N-terminal sequences contained a degenerate RXLR motif; whether or not this can function similarly to the canonical RXLR motif is yet to be determined. Functional studies such as those done for effectors from *Phytophthora* [51] would be informative to ascertain which of these degenerate motifs can function similarly to the canonical RXLR motif in protein secretion [16]. However, at present, the degenerate RXLR regular expression used in this paper should not be used on its own to search for genes encoding RXLR proteins due to the high false positive rate (>70%; Table 1). WY effectors lacking the non-canonical RXLR motif had features similar to proteins containing the canonical RXLR motif such as high N-terminal intrinsic disorder and the presence of an EER motif. N-terminal intrinsic disorder has been predicted for RXLR effectors of *Phytophthora* [46] and is also a common feature of bacterial effectors [52]. The biological significance of these features remains to be determined; however, it may be that these intrinsically disordered domains are important in post-translational modification or protein–protein interactions, as is the case for intrinsically disordered regions in other organisms [53,54].

The number of WY domains per protein varied considerably, from one to eight in *B. lactucae*. Many pathogen effectors exhibit rapid evolution and divergence due to the selective pressure of evolving host targets and host resistance proteins [26]. Duplication of domains may allow for evolution of novel effector functions or for evasion of host recognition while retaining function [31]. Variation in the number of repeated domains is reminiscent of NB-LRR proteins, which recognize effectors (or their activity) and also have leucine-rich repeat domains [55,56]. Duplication of regions encoding WY domains may have happened within a gene through replication errors or between different genes through recombination. The genomic sequences of the repeats can be analyzed in multiple isolates to reveal the origins of these duplications and domain expansion or contraction as has been done for NB-LRR proteins [57].

Analysis of the amino acid sequences of the WY domain in *B. lactucae* showed that it may be better considered a W[Y/F] domain, due to the equal preference for phenylalanine and tyrosine for the second characteristic residue. This substitution has a BLOSUM score of 3, indicating that it is fairly common. Both tyrosine and phenylalanine are aromatic amino acids; however, the hydroxyl group on tyrosine makes it slightly bulkier, more polar, and introduces a potential phosphorylation site. Variation in this region is not uncommon in other oomycete species: the fifth of the seven domains in Psr2 of *P. sojae* has an F instead of a Y and is similar in structure to the single domain of ATR1 in *H. arabidopsidis* that has a cysteine at the Y position [31]. Further structural characterization is needed to reveal whether these substitutions alter tertiary structure and biological function.

In oomycetes, WY domain containing effectors have been shown to have several functions including PTI suppression [58]. Here we show five WY effectors from *B. lactucae*, four of which lack the canonical RXLR motif, are able to suppress the host immune system by interfering with pathogen-triggered production of ROS. Suppression of host defenses is critical to the survival of *B. lactucae* and therefore it is not surprising that multiple effectors target the basal

immune system. Identification of the host targets of these effectors will determine which steps in the signal transduction cascade are modulated by each effector or may reveal candidates for susceptibility genes in the host that are required for successful proliferation of *B. lactucae*.

Effectors are powerful tools for the discovery and characterization of host resistance genes [59]. Eleven of the *B. lactucae* effectors tested caused necrosis or severe chlorosis on one or more lettuce lines. Necrosis and/or chlorosis is characteristic of recognition of effectors by R genes. This reaction could also be due to the non-specific toxic effects of effectors; however, the fact that necrosis only occurs in specific lettuce genotypes is consistent with a specific inter-action with R genes rather than a general toxicity. R genes that recognize effectors can be iden-tified by map-based cloning using segregating $F_{2:3}$ and RIL populations. Loci for recognition of five *B. lactucae* effectors (BLG01, BLG03, BLN08, BLR31, and BLR38) have been successfully mapped in other studies [24,45,60]. Two of these effectors, BLG01 and BLN08, have been shown to be recognized broadly by the non-host *L. saligna* [24,60]. Our study confirms the results for BLN08 and revealed BSW14 as an additional effector recognized by *L. saligna*. BLN08 and BSW14 share little sequence similarity though both contain WY domains. Non-host resistance in *L. saligna* is clearly complex [61], but these results indicate that it is mediated in part by recognition of multiple effectors.

Resistance to *B. lactucae* is present in some accessions of *L. virosa* [62,63]; however, its genetic basis and efficacy this has been little studied due to sexual incompatibility between *L. virosa* and other *Lactuca* spp. The lettuce lines ViAE and ViCQ contain introgressed disease resistance from *L. virosa*, which was achieved using embryo rescue followed by extensive back-crossing to *L. sativa* [49]. We have shown that ViAE and ViCQ recognize five WY domain containing effectors from *B. lactucae* and are therefore expected to be highly resistant to *B. lac-tucae* in the field; this is consistent with resistance of ViAE to all 162 isolates tested from Cali-fornia (https://bremia.ucdavis.edu/bremia_database.php). For ViAE, we show that these effector recognitions are genetically linked to each other, and importantly, completely geneti-cally cosegregate with resistance to *B. lactucae*. Experiments are underway to determine the molecular basis of this resistance.

The non-RXLR, WY containing effector BSW04p is a candidate for Avr11 due to recogni-tion by multiple lettuce lines containing *Dm11*. Confirmation that *Avr11* encodes BSW04p requires assessing the nucleotide and expression level polymorphism of the effector in virulent and avirulent isolates of *B. lactucae*, as well as cloning the cognate R gene and testing whether it confers resistance to isolates expressing Avr11. Interestingly, the *Dm11* containing lines that recognize BSW04p do not recognize the paralog BSW04m, which elicits cell death in other lines of lettuce, such as ViAE and one accession of *L. serriola*. Using domain swap experiments, we narrowed down the region responsible for cell death in Ninja to the C-terminal half of BSW04p, in a region located in between two WY domains; because WY proteins often contain multiple equally spaced WY repeats, it is possible that this region also contains a WY domain that was not predicted by the HMM [31]. Structural characterization is needed to determine whether this region forms a WY-like fold and what structural consequences the amino acid differences have between BSW04p and BSW04m.

Not all RXLR candidate effectors have WY domains and the presence of a WY domain is not required for the avirulence phenotype of all effectors; for example, WY domains were not predicted for *B. lactucae* effectors BLG01 and BLG03, yet they elicit an immune response in lettuce [24]. Structural elucidation of an RXLR effector lacking the WY motif *H. arabidopsidis* ATR13 revealed that it contained a helical fold that was distinct from the WY fold [64]. It would be informative to identify and compare the protein structures of additional RXLR effec-tors that lack WY domains to determine whether there are other conserved C-terminal domains that may be involved in effector functions.

RXLR and WY effectors provide tools for monitoring pathogen populations and effector-driven resistance breeding. Analysis of phenotypically diverse isolates from multiple locations will allow the characterization of individual effector repertoires as well as the development of the pan-repertoire for the pathogen species. Cloned effectors also will be highly instrumental for identification of their cognate resistance genes as well as effector targets in the host. In addition, screens for resistance using transient expression of individual effectors will allow the pyramiding of resistance genes with different specificities that will maximize the evolutionary hurdle for the pathogen to become virulent. Ultimately, knowledge of effector repertoires will allow data-driven deployment of resistance genes leading to more durable disease resistance [65].

## Materials and methods

### Effector prediction

**Signal peptide and WY domain prediction.** To search for the WY domain, an HMM was built using HMMer v3.1 [66] based on 721 amino acid sequences of WY domains in RXLR motif containing proteins from *P. infestans*, *P. soja*e, and *P. ramorum* obtained from Boutemy *et al.* [28]. The sequence described by the HMM spans the WY domain in the crystal structures of Avr3a11 and PexRD2 [27]. Candidate WY effectors were predicted using this HMM to search translated predicted ORF sequences (>80 amino acids) as well as gene models in the draft genome of *B. lactucae* isolate SF5 [44]. Sequences with a positive HMM bit score were considered to be putative WY domain effectors as in [28]. We also used an HMM built from predicted *B. lactucae* WY domains, but found that it did not identify any additional candidates compared to the *Phytophthora* based HMM. Signal peptide prediction was performed on candidate WY effectors using SignalP v 4.1 [67] and PhobiusSP [68]. Default settings were altered for SignalP v 4.1 to have sensitivity similar to SignalP v 3.0. Output was compared between SignalP v 4.1 sensitive and the combination of SignalP4.0 + SignalP 3.0, and identical results were obtained from the two methods. Gene models were more accurate for predicting signal peptides than translated ORFs—in part due to misannotated start codons upstream of the probable true start codon and signal peptides in the ORFs. However, on several occasions the gene model was missing a signal peptide found in the ORF; these gene models were manually updated to reflect this likely true start site. SignalP v 4.1 in sensitive mode was better able to predict signal peptides in proteins with misannotated start codons compared to SignalP v 4.0 or v 5.0.

**RXLR and EER prediction.** A combination of string searches and HMMs were used to search for the RXLR motif in oomycete predicted WY proteins. The following strings were used based on variants observed in downy mildews: [RQGH]XLR for RXLR and [DE][DE][KR] for EER. An HMM from Whisson *et al.* [15] was also tested, although this did not reveal any additional RXLR motifs. To search for additional non-canonical RXLR motifs in WY candidates, a highly degenerate string of [RKHGQ][X]{0,1}[LMYFIV][RNK] was used. Searches were limited to the first 100 amino acids of the N-terminus of the predicted protein sequence.

**Estimation of false positive rates for effector prediction.** We determined false positive rates for each motif (Table 1) by analyzing multiple permutations of the non-redundant secretome using the same pipeline as described above for effector prediction. At least 10 random permutations of the sequence space were created using the MEME fasta-shuffle-letters program (with a kmer size of 1) with peptides starting after the cleavage site identified by SignalP v4.1. The false positive rate for each motif was estimated as the average frequency of detection in the permutated sequences divided by the observed frequency in the original sequences.

## Sequence comparison

To generate a neighbor-joining tree of the WY effector candidates, whole protein amino acid alignments were performed using MUSCLE 3.8.425 [69] implemented in Geneious 11.0.5 (http://www.geneious.com) with default settings. A UPGMA tree was built in Geneious with bootstrap resampling (100 replicates). The resulting tree was annotated using the Interactive Tree of Life [70] and stylistically refined in the GNU Image Manipulation Program (http://www.gimp.org). To build a sequence logo for the WY domain from *B. lactucae*, a multiple sequence alignment of WY domains was built using MUSCLE 3.8.425 [69], manually corrected for alignment errors, and the sequence logo generated using Weblogo (http://weblogo.berkeley.edu/logo.cgi).

## Prediction of intrinsic disorder

PONDR VSL2 [71] was used to calculate levels of intrinsic disorder for groups of candidate effectors, grouped by the presence of an RXLR-like motif, EER-like motif, and/or WY domain. PONDR VSL2 is a meta-predictor of protein disorder that utilizes neural networks and amino acid context to give a weighted disorder score at each amino acid position. The average sequence disorder was calculated at each amino acid position in each group of effectors and aligned on the first amino acid. Plots were generated from the average positional disorder scores calculated from all peptides in a group. The entire non-redundant, predicted secretome was used as a reference for comparison.

## Lineage specificity

To determine if the candidate WY proteins are unique to *B. lactucae*, BLASTp [72] was used to search for orthologs in other oomycete species. BLASTp-based sequence comparisons with an e-value threshold of 0.01 were performed against the following oomycete genomes: *Albugo laibachii* [8], *H. arabidopsidis* [73], *P. capsici* [74], *Phytophthora cinnamomi* [75], *P. infestans* [3], *Phytophthora parasitica* [76], *P. ramorum* [5], *P. sojae* [5], *Pythium ultimum* [7], *Saprolegnia parasitica* [77], *Pseudoperonospora cubensis* [22], *Plasmopara halstedii* [25], *Plasmopara viticola* JL-7-2 [78] and INRA-PV221 [79], *Peronospora tabacina* [80], and *Peronospora effusa* [81].

## Gateway cloning of effector candidates

Twenty-one WY effectors that differed in their number of WY domains were selected at random for cloning and functional analysis. Genes were cloned from amplified genomic DNA in an attempt to capture both alleles of each effector from the homokaryotic isolate SF5 [44]. In order to capture additional allelic diversity, some effectors were also cloned from the heterokaryotic isolate C82P24 [44]. Effector candidates without their signal peptide were cloned into the pEarleyGate100 (pEG100) vector for plant expression [82] using Gateway cloning (Thermo Fisher Scientific). Candidate effector genes were amplified from genomic DNA isolated from spores of *B. lactucae* isolate SF5 or C82P24 using Phusion High Fidelity Polymerase (Thermo Fisher Scientific) and primers (S4 File) that amplified each ORF after the predicted signal peptide cleavage site. The Kozak sequence (ACCATG) was added to the forward primer for correct translational initiation. PCR products were purified using precipitation with either polyethylene glycol (PEG) or homemade solid phase reversible immobilization (SPRI) beads to remove primers and primer-dimers, recombined into pDONR207 using BP Clonase II (Thermo Fisher Scientific), and transformed into chemically competent *Escherichia coli* DH10B cells. The resulting entry clones were sequenced using primers designed to

pDONR207 to confirm gene identity and identify alleles. Two alleles were obtained for many effectors; in addition, more than two alleles were obtained for several genes with the additional alleles representing chimeras between the two haplotypes of isolate SF5, which is diploid and homokaryotic. The chimeric sequences show distinct recombination breakpoints and may have arisen either during PCR or Gateway-based recombination. Clones of all unique sequences for each effector were retained because they could be informative for dissecting the sequence basis of host recognition. This resulted in multiple distinct clones (wildtype alleles plus chimeric sequences) for some effectors.

The entry clones were then recombined into pEG100 using LR Clonase II and transformed into *E. coli* to generate expression clones. The expression clones were then transformed into *Agrobacterium tumefaciens* strain C58rif⁺ using electroporation. Kanamycin-resistant colonies were confirmed for the transgene using gene-specific primers. For the domain-swap experiments, chimeras were generated using PCR-fusion/Gateway cloning using overlapping primers as previously described [83]. For localization experiments, effectors were cloned without their signal peptide either as described above into pEG104 [31], resulting in an N-terminal YFP fusion, or they were cloned into the Gateway compatible binary vector pl2B-KAN-LjUB 1-GW-tHSP with mCherry added using overlapping primers [83]. The vector pl2B-KAN-L jUB1-GW-tHSP was constructed using Golden Gate assembly [84] with EC50507 T-DNA backbone vector (https://www.ensa.ac.uk/resources/) with a Gateway acceptor site added (including ccdb) and a pNOS-Kan-tNOS cassette added.

## Gene expression analysis

Expression of the candidate WY effectors in *B. lactucae* was evaluated using RNA-seq reads from a seven-day timecourse experiment of *B. lactucae* infecting lettuce seedlings (https://www.ncbi.nlm.nih.gov/bioproject/PRJNA523226). Reads were mapped to all gene models using SAMtools [85] and then the candidate WY gene models were extracted and counted using BEDtools multicov [86]. Raw read counts are given in S1 File. To confirm expression of all cloned candidate WY effectors, specific quantitative real-time PCR (qRT-PCR) primers were designed for each transcript for SYBR Green based quantification [87]. Five seedlings per replicate of *L. sativa* cv. Cobham Green infected with *B. lactucae* isolate SF5 were collected and frozen at 0, 2, 4, and 6 days post infection with uninfected seedlings also taken as a negative control. Total RNA was extracted from seedlings using the RNeasy Plant Kit (Qiagen). Samples were treated with DNase to remove genomic DNA and RNA was reverse transcribed into cDNA using RevertAid RT (Thermo Fisher Scientific). qRT-PCR was performed using PerfeCTa SYBR Green SuperMix (Quantabio) on a 7900HT Fast Real-Time PCR machine (Applied Biosystems).

## *Agrobacterium*-mediated transient assays

Agroinfiltration and transient expression experiments were performed on *N. benthamiana* and lettuce using conditions optimized for lettuce [88]. Cultures of *A. tumefaciens* strain C58Rif+ were started from glycerol stocks in Luria-Bertani (LB) media supplemented with 50 μg/μL each rifamycin and kanamycin and grown for 16–18 hours at 30°C. For infiltration, cultures were pelleted, washed, and resuspended in 10 mM MgCl₂ to OD = 0.3–0.5.

## Subcellular localization prediction and characterization

Subcellular localization was assessed for nine cloned candidate WY effectors using Agroinfiltration as described above. Three to five days post-Agroinfiltration, lettuce leaves expressing N-terminal YFP or mCherry fusions of effectors were examined for subcellular localization by

confocal microscopy. Nuclear staining was performed by incubating cut leaf tissue in 18 nM DAPI in water for at least five minutes. Imaging was performed on a Zeiss LSM 710 laser scanning confocal microscope with a 40x objective lens. Nuclear localization was predicted using NucPred [47] and nuclear localization motifs were predicted using LOCALIZER [89].

## Western blotting

Leaf tissue was collected from lettuce or *N. benthamiana* at 2–3 days post infiltration. Proteins were extracted from leaf tissue using GTEN buffer (10% glycerol, 25 mM Tris pH 7.5, 1 mM EDTA, 150 mM NaCl) supplemented with 2% PVPP, 10 mM DTT, 1x protease inhibitor cocktail, and 0.1% Tween-20 [90]. Total protein (20 μg) was run on a 4–12% Bis-Tris SDS-PAGE gel at 200V for 30 minutes using MOPS SDS running buffer. Proteins were transferred to a PVDF membrane using wet transfer and were probed using anti-GFP (Roche 1181446001) or anti-mCherry (BioVision 5993–100) antibodies. Secondary HRP-conjugated antibodies were used for visualization on x-ray film using ECL reagent (Pierce).

## PTI suppression assay

Effectors were expressed in five-week-old plants of *N. benthamiana* using Agroinfiltration as described above. Two days post-infiltration, two leaf discs (3.8 mm) were taken using a cork borer from each infiltration site away from leaf veins. Leaf discs were floated abaxial side up in 200 μL of distilled water in a 96-well white assay plate and incubated at room temperature for 24 hrs. To measure suppression of ROS production [91], the water was removed and 100 μL of assay solution was added to the leaf discs, which contained 17 μg/mL luminol (Sigma-Aldrich, St Louis, MO) and 10 μg/mL horseradish peroxidase type 6A (Sigma-Aldrich). One of the two paired leaf discs was exposed to 100 nM of flg22 peptide (flg+) and the other was not exposed to any elicitor as a control for endogenous ROS production. Three published bacterial suppressors of PTI from *P. syringae* pv *tomato* DC3000 [92] were used as positive controls for suppression of ROS production: HopS2 [93], HopX1, and HopT1 [94]; GFP and DsRed were used as negative controls. Luminescence was measured on a FilterMax F5 Plate Reader (Molecular Devices, San Jose, CA) promptly after adding in the assay solution and was measured every two minutes for a total of 40 minutes. For the untagged proteins, the assay was repeated on two separate occasions with a total of 15 replicates for the flg22+ treatment and eight replicates of the flg22- treatment. For the fluorescently-tagged proteins, the assay was repeated three times with a total of 16 replicates for both the flg22+ and the flg22- treatments. For quantification of total ROS burst, the area under the curve for luminescence was calculated from 2–40 minutes in R version 3.5.1 [95] using the 'auc' function with liner interpolation in the MESS package [96].

## Lettuce germplasm screen for effector recognition

Agroinfiltration was performed on lettuce as described above. The youngest fully expanded leaves of 3 to 4-week-old greenhouse grown lettuce plants were infiltrated with the *A. tumefaciens* cultures using a needleless syringe. Leaves were examined four to five days post-infiltration for signs of macroscopic cell death, indicative of immune recognition by host resistance proteins. *A. tumefaciens* containing pEG100:35S-GFP was used as a negative control; *A. tumefaciens* containing pBAV139:35S-HopM1 from *Pseudomonas syringae* [92] or pB7WG2:35S-PsojNIP from *P. sojae* [97] were used as positive controls. To control for false negatives, leaves that did not show cell death in response to the positive control were excluded from the analysis. To control for false positives, leaves that showed cell death in response to GFP control were also excluded from the analysis. In addition, any cell death observed in the

initial screening was confirmed by repeating infiltrations on at least four plants for effectors found to cause cell death.

### *B. lactucae* disease assays

Lettuce seedlings were screened for resistance or susceptibility to *B. lactucae* as described previously [49]. Briefly, cotyledons of week-old lettuce seedlings were sprayed with spores from *B. lactucae* isolate Bl: 23 at a concentration of $1–2 \times 10^5$ spores/mL and scored for presence/absence of sporulation at 7 and 14 days post inoculation.

### Supporting information

**S1 Table. RXLR-like sequences observed within 100 amino acids of the N-terminus of *B. lactucae* WY proteins, sorted by first amino acid.** Sequences that occur in more than one protein are highlighted in red.
(DOCX)

**S2 Table. Number of predicted secreted WY effectors with or without RXLR and/or EER motifs in the genomes of downy mildew pathogens and *Phytophthora* species.**
(DOCX)

**S1 Fig. Quantitative real-time PCR results for candidate effectors from *B. lactucae*. (A)** Normalized expression levels for each effector transcript at 0, 2, 4, and 6 days post infection (dpi) and uninfected lettuce as a control. *B. lactucae* B-tubulin (BlBtub) was used as a housekeeping gene to quantify total amount of *B. lactucae* RNA on infected seedlings and *L. sativa* actin (LsAct) was used to quantify the amount of lettuce RNA. Transcript levels normalized to B-tubulin using the $2^{\Delta Ct}$ method ($2^{(Ct\ effector–Ct\ b-tubulin)}$)[98]. **(B)** Raw Ct values for each effector transcript at 0, 2, 4, and 6 days post infection (dpi) and uninfected lettuce as a control. *B. lactucae* B-tubulin (BlBtub) was used as a housekeeping gene to quantify total amount of *B. lactucae* RNA on infected seedlings and *L. sativa* actin (LsAct) was used to quantify the amount of lettuce RNA.
(TIFF)

**S2 Fig. Intrinsic disorder of first 150 amino acids of RXLR and WY containing candidate effectors predicted from other oomycete genomes.** The x-axis shows the amino acid position; the y-axis shows the intrinsic disorder (VSL2 PONDR) score. Proteins were categorized as WY with no RXLR (yellow), WYs with RXLR (black), RXLR+EER (with or without WY, purple), and the total predicted secretome (blue) using the same color scheme as Fig 2. RXLR and EER motifs were as described in Figs 2 and 3. Average positional disorder scores are shown for each class of proteins.
(TIFF)

**S3 Fig. Stability of fluorescently tagged WY effectors transiently expressed in lettuce. (A)** Anti-mCherry (mCh) Western blot. **(B)** Anti-YFP Western blot. **(C)** Expected sizes for each of the fusion proteins.
(TIFF)

**S4 Fig. Stability of fluorescently tagged WY effectors transiently expressed in *Nicotiana benthamiana*. (A)** Anti-mCherry (mCh) Western blot. **(B)** Anti-YFP Western blot. **(C)** Expected sizes for each of the fusion proteins.
(TIFF)

**S5 Fig. Agroinfiltration results from BLN06-SF5 and sequence comparison between BLN06, SF5, and BL24. (A)** Agroinfiltration scores from various isoforms of BLN06 from *B. lactucae* isolate SF5. Scores as in Fig 7. One leaf per plant was infiltrated and scores were averaged across 10 replicates. All leaves were confirmed to be expressing mCherry (mCh)-BLN06 by confocal microscopy. **(B)** Example confocal microscopy of N-terminal mCherry-tagged BLN06 in LS102, RYZ2164, and NunDm17 lettuce genotypes. **(C)** Sequence comparison between BLN06 isoforms from isolates SF5 and BL24 [45]. BLN06 was initially cloned from SF5 with the first 28 amino acids removed in case there was a cryptic signal peptide. For the mCherry-BLN06 fusions, these 28 amino acids were included in the protein sequence. Both alleles of BLN06 were cloned from SF5 (BLN06-SF5-1 and BLN06-SF5-2).
(TIFF)

**S6 Fig. BSW04m is expressed in Ninja but does not cause cell death. (A)** Agroinfiltration scores from *L. sativa* cv Ninja three days post infiltration. GFP is used as a negative control, HopM1 is used as a positive control for necrosis. **(B)** Representative photo of infiltrated Ninja leaves before clearing (left) and after clearing (right) with ethanol to visualize necrosis (brown areas). **(C)** Example confocal microscopy result of mCherry-BSW04m (red) transiently expressed in Ninja. All four leaves showed similar expression and nuclear localization. Chlorophyll autofluorescence is shown in blue. The fusion protein is intact as measured by Western blot (S3 Fig).
(TIFF)

**S1 File. Sequences and NCBI reference numbers of the 59 WY proteins predicted from the *B. lactucae* SF5 genome assembly.**
(XLSX)

**S2 File. Sequences of predicted translated ORFs from 14 oomycete genomes with results from RXLR and EER motif searches and WY HMM searches on each sequence.**
(XLSX)

**S3 File. Lettuce genotypes tested for recognition of *B. lactucae* effectors.**
(XLSX)

**S4 File. Primers used in Gateway Cloning of the *B. lactucae* effectors and for qRT-PCR.**
(XLSX)

## Acknowledgments

We thank Keri Cavanaugh, Alyssa Schweickert, Catherine Lopez, Amber Robbins, Natalie Hamada, Jesus Cardenas, Pauline Sanders, and Sebastian Reyes Chin Wo (all UC Davis) for wet lab and computational assistance. We thank Anne Giesbers for the plasmid pl2B-KAN-LjUB1-GW-tHSP. We thank Gitta Coaker (UC Davis) for assistance with the ROS assay. We thank Ales Lebeda (Palacký University, Olomouc, Czech Republic) and Alex Beharav (The Hebrew University, Tel Aviv, Israel) for lines of *L. saligna*. We thank Lien Bertier, Anne Giesbers, and Elizabeth Georgian for helpful comments on earlier versions of this manuscript.

## Author Contributions

**Conceptualization:** Kelsey J. Wood, Richard Michelmore.

**Formal analysis:** Kelsey J. Wood, Munir Nur, Juliana Gil, Kyle Fletcher, Kim Lakeman, Ayumi Gothberg, Tina Khuu, Jennifer Kopetzky, Archana Pandya.

**Funding acquisition:** Kelsey J. Wood, Richard Michelmore.

**Investigation:** Kelsey J. Wood, Munir Nur, Juliana Gil, Kim Lakeman, Dasan Gann, Ayumi Gothberg, Tina Khuu, Sanye Naqvi, Archana Pandya, Chi Zhang, Brigitte Maisonneuve.

**Supervision:** Kelsey J. Wood, Mathieu Pel, Richard Michelmore.

**Visualization:** Kelsey J. Wood.

**Writing – original draft:** Kelsey J. Wood, Richard Michelmore.

**Writing – review & editing:** Kelsey J. Wood, Richard Michelmore.

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
