## [Decision Letter · Decision Letter 0]

22 Jul 2020

Dear Prof. Michelmore,

Thank you very much for submitting your manuscript "Effector prediction and characterization in the oomycete pathogen Bremia lactucae reveal host-recognized WY domain proteins that lack the canonical RXLR motif" for consideration at PLOS Pathogens. As with all papers reviewed by the journal, your manuscript was reviewed by members of the editorial board and by several independent reviewers. In light of the reviews (below this email), we would like to invite the resubmission of a significantly-revised version that takes into account the reviewers' comments.

The Editors and reviewers all recognize the efforts the authors have undertaken to prepare a revised (resubmitted) version of the original manuscript, which is improved. However, there are outstanding comments from the reviewers that require attention. Of particular note, 2 reviewers request that Western Blots are performed to confirm the presence of expressed, intact proteins (critical for the interpretation of Figs 5, 8, and 10). These are also important for Fig 6, where variability in ROS suppression occurs with protein fusions that maybe related to protein abundance; my suggestion here is to select an example to confirm this hypothesis. Further, Figure 6 should be presented as a box plot a kin to Fig 4., this would help alleviate the concern of Reviewer 2 regarding the repeated HopS2 experiments. If the authors require additional time for these experiments (given the current situation), please request this. Please also add text to the manuscript to clarify reviewer comments such as those from reviewer 3 on the fact that cell death does not necessarily mean resistance.

We cannot make any decision about publication until we have seen the revised manuscript and your response to the reviewers' comments. Your revised manuscript is also likely to be sent to reviewers for further evaluation.

Sincerely,

Mark J Banfield

Guest Editor

PLOS Pathogens

Bart Thomma

Section Editor

PLOS Pathogens

Kasturi Haldar

Editor-in-Chief

PLOS Pathogens

orcid.org/0000-0001-5065-158X

Michael Malim

Editor-in-Chief

PLOS Pathogens

orcid.org/0000-0002-7699-2064

Reviewer's Responses to Questions

**Part I - Summary**

Reviewer #1: I reviewed the first submission of this manuscript (Reviewer 2). I am very pleased with the addition of Table S2, which clearly illustrates the magnitude of WY-based searches for identifying effector candidates that RXLR-based searches might miss. I agree with the authors' view that such differences are "significant".

I am also happy to see the new data from screens of the RILs, demonstrating cosegregation of effector-induced cell death and downy mildew resistance. These data buttress an already-strong case, from functional studies, that at least some of the candidates comprise bona-fide effectors.

At the time of the first submission, I was very enthusiastic about the significance of this work, and I remain so for the reasons described in my first review. I do not believe that time elapsed since the first submission has diminished the impact of this work; Indeed, the downloads of the manuscript from BioRxiv attest to the interest in this work. In sum, I am convinced that the advances in this manuscript will significantly improve the predictive value of bioinformatic pipelines to identify oomycete effectors. This is a substantial step forward for foundational understanding of plant-oomycete interactions and for translational research to improve breeding for oomycete disease resistance.

Reviewer #2: I revised a previous version of the manuscript. The new version presents new results, in particular the results obtained with the RILs, that reinforce the idea that some WY-domain containing proteins are being recognized by R genes and thus they must be translocated inside plant cells upon infection. It is however unfortunate that the authors could not perform wester-blots due to the current sanitary situation, because showing that the different proteins are being produced at the same levels is an important element to support some of the conclusions of the paper, in particular those involving differential recognition of putative effectors by different lettuce genotypes, as well as those involving domain-swap experiments.

Considering novelty, since the first submission of the manuscript there has been a publication reporting the identification of WY-domain proteins lacking RXLR motifs in a different downy mildew species (Combier et al, PLoS ONE, 14(7): e0220184). This paper should be cited, at least in the introduction.

Reviewer #3: The authors present bioinformatic prediction of candidate WY-containing effectors. It is disappointing that they have removed data indicating whether the effector candidates are up regulated during infection as this is a key criterion giving confidence to the assertion that they are indeed effectors. The work on localisation and PTI suppression remains preliminary and unconvincing.

The triggering of cell death does not mean that an effector has been ‘recognised’ by a host resistance. It could also be because overexpression of the effector has become toxic to the plant cell. This can occur, for example, when excess effector could have off-target effects, or could prevent basic host cell functions, such as normal membrane integrity or fusions. Interestingly, overexpression of BSW14, for example, causes cell death in all saligna genotypes.

The addition of Figure 9 data reveals strong evidence that avirulences have been found for sativa, including BSW14.

**Part II – Major Issues: Key Experiments Required for Acceptance**

Reviewer #1: None. As noted above, I'm quite satisfied with the responses to the reviews.

Reviewer #2: Unfortunately, Western-Blots are required to validate some of the manuscript conclusions, in particular those issued from results presented in Figures 8 and 10.

I am puzzled by the fact that in the first version of the manuscript HopS2 did not suppress the flg22-mediated ROS burst and in this new version it does? How do the authors explain this?

Figure 10: I may have missed something, but based on the results presented in the manuscript I don’t understand how the authors narrow down the recognized region to residues 405-458 (there seem to be lots of aa differences C-terminal to this region).

Figure S3: I still believe that BLN06 from BL24 should be included in this experiment in order to show that it is recognized by the different lettuce lines in the experimental conditions used by the authors.

Reviewer #3: Regarding the previous request to provide western data confirming that effector-FPfusion proteins were stably expressed, I agree that this is not required for all 30,000 in planta expressions. And I sympathise with situations where labs have struggled to perform lab work due to the pandemic. It was not a frivolous request for new data it was a request for basic experiments without which conclusions cannot be made. The localisations in Figure 5 may, or may not, be due to full-length effector-FP fusions. They could be due to cleavage products, i.e. the FP is clearly intact but how much of the effector is present? Or, where the localisation is the same as you would expect for FP-only controls (which are missing), how do we know that any effector is present? For me, this Figure adds nothing. For sure, we expect effectors to occupy different subcellular locations, but I can’t conclude anything from the Figure as it stands. In Figure 6 the authors indicate that PTI suppression activity of effectors can be altered by the addition of an FP fusion. The ability of the untagged BSW14 and BSW19 to suppress PTI is lost by the FP-fusion versions. In contrast, untagged BSW04p and BSW13 fail to suppress PTI, whereas the tagged versions do. Further, the ability of untagged BSW03 to enhance the ROS burst (an unusual effector activity) is prevented by addition of the FP fusion. Are all of these fusions intact? If not, what do we make of the localisation data in Figure 5?

**Part III – Minor Issues: Editorial and Data Presentation Modifications**

Reviewer #1: Fig. 2 and Table S2: The category labelled "RXLR-EER (no WY)" should labelled as "RXLR+EER (no WY)"

Might it be possible to add a supplementary table with the specific IDs of genes in various categories from Fig. 2/TableS2? I recognize that such an addition does not bear on the strength of support for the main points of the manuscript, but would be a useful resources for effector biologists to identify potentially interesting effectors without having to re-do the searches that this manuscript describes. To be clear: I am not suggesting that such an addition is necessary for acceptance. I consider it optional, but I hope that the authors will take this extra step as a means of amplifying the impact of their nice work.

Reviewer #2: Figure 5: I find it difficult to define the subcellular localization for BSW14 and BSW19 based on the provided images.

Figure 7: compared to the previous version of the manuscript the figure has lost an inset explaining the different color codes which was very useful to interpret the figure.

Legend of Figure 8 refers to “magenta text” and “aqua text”, but the colors are not present in the figure.

Figure 9: while it is understandable that in Figure 7 only average scores are reported for presentation reasons, here it will be suitable to present also information on dispersion, ideally showing all data as box plots. Also, the Figure refers to 67 RILs but the text states 70 RILS (l. 293). Based on the scores shown for the combination BSW14-ViAE in Figures 7 and 9 there seems to exist a variability that justifies presenting the dispersion.

Legends of Figures 9 and 10 refer to Figure 6 but it is actually Figure 7.

Figure 10: understanding the composition of the different chimeric proteins obtained by domain swaps is difficult; a simple diagram using color-coded boxes will improve the comprehension of the figure. In section C, from the formal point of view it is surprising to see protein domains and aminoacid changes positioned on a nucleotide sequence.

Figure S1: the legend is inverted for A and B. Also, it would be nice if the authors explained how the transcript levels are calculated, or gave a reference for the method.

Discussion, l. 436: if I understand properly, the authors mean BSW04m. Concerning the sentence itself, ideally it should be shown that the protein is expressed in the lettuce line where it does not elicit cell death (there could be differences of expression between lettuce lines).

Reviewer #3: (No Response)

PLOS authors have the option to publish the peer review history of their article (what does this mean?). If published, this will include your full peer review and any attached files.

Reviewer #1: **Yes: **John M. McDowell

Reviewer #2: No

Reviewer #3: No
---

## [Editor Report · Decision Letter 1]

29 Sep 2020

Dear Prof. Michelmore,

We are pleased to inform you that your manuscript 'Effector prediction and characterization in the oomycete pathogen Bremia lactucae reveal host-recognized WY domain proteins that lack the canonical  RXLR motif' has been provisionally accepted for publication in PLOS Pathogens.

Best regards,

Mark J Banfield

Guest Editor

PLOS Pathogens

Bart Thomma

Section Editor

PLOS Pathogens

Kasturi Haldar

Editor-in-Chief

PLOS Pathogens

orcid.org/0000-0001-5065-158X

Michael Malim

Editor-in-Chief

PLOS Pathogens

orcid.org/0000-0002-7699-2064
---

## [Editor Report · Acceptance letter]

16 Oct 2020

Dear Prof. Michelmore,

We are delighted to inform you that your manuscript, "Effector prediction and characterization in the oomycete pathogen *Bremia lactucae* reveal host-recognized WY domain proteins that lack the canonical  RXLR motif," has been formally accepted for publication in PLOS Pathogens.

Best regards,

Kasturi Haldar

Editor-in-Chief

PLOS Pathogens

orcid.org/0000-0001-5065-158X

Michael Malim

Editor-in-Chief

PLOS Pathogens

orcid.org/0000-0002-7699-2064